# Interoperability of RTN1A in dendrite dynamics and immune functions in human Langerhans cells

**Małgorzata Anna Cichoń[1], Karin Pfisterer[1], Judith Leitner[2], Lena Wagner[1], Clement Staud[3], Peter Steinberger[2], Adelheid Elbe-Bürger[1]***

[1]Department of Dermatology, Medical University of Vienna, Vienna, Austria; [2]Center for Pathophysiology, Infectiology and Immunology, Medical University of Vienna, Vienna, Austria; [3]Department of Plastic and Reconstructive Surgery, Medical University of Vienna, Vienna, Austria

**\*For correspondence:**
adelheid.elbe-buerger@
meduniwien.ac.at

**Competing interest:** The authors declare that no competing interests exist.

**Abstract** Skin is an active immune organ where professional antigen-presenting cells such as epidermal Langerhans cells (LCs) link innate and adaptive immune responses. While Reticulon 1A (RTN1A) was recently identified in LCs and dendritic cells in cutaneous and lymphoid tissues of humans and mice, its function is still unclear. Here, we studied the involvement of this protein in cytoskeletal remodeling and immune responses toward pathogens by stimulation of Toll-like receptors (TLRs) in resident LCs (rLCs) and emigrated LCs (eLCs) in human epidermis ex vivo and in a transgenic THP-1 RTN1A[+] cell line. Hampering RTN1A functionality through an inhibitory antibody induced significant dendrite retraction of rLCs and inhibited their emigration. Similarly, expression of RTN1A in THP-1 cells significantly altered their morphology, enhanced aggregation potential, and inhibited the $Ca^{2+}$ flux. Differentiated THP-1 RTN1A[+] macrophages exhibited long cell protrusions and a larger cell body size in comparison to wild-type cells. Further, stimulation of epidermal sheets with bacterial lipoproteins (TLR1/2 and TLR2 agonists) and single-stranded RNA (TLR7 agonist) resulted in the formation of substantial clusters of rLCs and a significant decrease of RTN1A expression in eLCs. Together, our data indicate involvement of RTN1A in dendrite dynamics and structural plasticity of primary LCs. Moreover, we discovered a relation between activation of TLRs, clustering of LCs, and downregulation of RTN1A within the epidermis, thus indicating an important role of RTN1A in LC residency and maintaining tissue homeostasis.

## Editor's evaluation

This is an important manuscript that establishes a novel link between a novel molecule named Reticulon 1A expressed in Human Langerhans cells and their important immune functions such as migration to lymph nodes and T cell activation. The data presented are novel and convincing. Notably, the authors present intriguing parallels with the mechanisms that control dendrite growth in neurons which may also be relevant for migrating dendritic cells. Altogether, this manuscript will be of wide interest to the scientific community working in the fields of immunology, vaccinology, dermatology, cell biology, and beyond.

## Introduction

A well-functioning network of dendritic cells (DCs) and macrophages (Mφs) in the upper layer of human skin is of great importance to defeat invading pathogens and to sustain the intact skin barrier. Langerhans cells (LCs) reside in the epidermis in an immature state, have features of both DCs and

Mφs, and can function in maintaining tolerance but also priming an immune response (*Eisenbarth, 2019*; *Mass et al., 2016*; *Doebel et al., 2017*). Recently, more light was shed on LC diversity and how the phenotype of LC subtypes can vary depending on the function they are dedicated to *Shibaki et al., 1995*; *Collin and Bigley, 2021*; *Liu et al., 2021*. Moreover, in healthy skin the LC fate can be shaped by the environment (*Deckers et al., 2018*; *West and Bennett, 2017*; *Clayton et al., 2017*). Responsiveness to external intruders results in activation and maturation of certain LC subsets sensing antigens via activation of Toll-like receptors (TLRs), a category of pattern recognition receptors that initiate the innate immune response. They not only recognize specific microbial particles termed pathogen-associated molecular patterns including lipopolysaccharides of gram-negative and lipoteichoic acid of gram-positive bacteria, and nucleic acids of viruses, but also endogenous damage-associated molecular pattern molecules. TLRs which recognize nucleic acids reside in intracellular compartments to decrease the risk to encounter 'self' nucleic acids, whereas cell surface TLRs largely recognize microbial membrane compartments and therefore do not require this protective strategy (*van der Aar et al., 2007*; *van der Aar et al., 2013*; *Delgado and Deretic, 2009*; *Flacher et al., 2006*; *Kawai and Akira, 2011*; *Peiser et al., 2008*; *Tajpara et al., 2018*). Activated LCs downregulate molecules necessary for tissue exit such as E-cadherin (*Jiang et al., 2007*; *Riedl et al., 2000*), upregulate co-stimulatory molecules, secrete cytokines, and undergo a complex cytoskeleton remodeling that enables LCs to acquire a roundish shape with short dendrites to efficiently relocate within and through the epidermis toward dermal lymphatic vessels and with these short dendrites having transformed into thin cytoplasmic protrusions ('veils'), lastly toward lymph nodes (*Randolph et al., 2005*). The crosstalk between LC activation and morphological changes is not fully understood. LCs can be distinguished by their morphological plasticity, namely their readily dendrite extension during their residence and replenishment in the epidermis (*Elbe et al., 1989*; *Geissmann et al., 2010*) and changes in dendrite patterning resulting in dendrite retraction due to their intra-tissue migration capacity (*Hunger et al., 2004*). LC dendrites are membranous extensions, which in healthy skin exhibit orderly dendrite distribution with small distance to other cells (*Park et al., 2021*). Notably, dendrites can display similar structures, branching, and intersection frequency as neurons (*Öztürk et al., 2020*).

The endoplasmic reticulum (ER)-associated protein Reticulon 1A (RTN1A) was recently identified in DCs of cutaneous and lymphoid tissues in humans and mice (*Gschwandtner et al., 2018*; *Cichoń et al., 2020*). Previously, an interesting observation was made on the distribution of the RTN1A protein within LC dendrites (*Cichoń et al., 2020*). Indeed, some ER-associated proteins (e.g. Atlastin-1) have been shown to be involved in the distribution and shaping ER morphology in neural dendrites (*Liu et al., 2019*). Certain RTN family members have been associated with ER morphogenesis (*Hu et al., 2009*; *Voeltz et al., 2006*). In neuroanatomy research, the involvement of RTNs in neurite expansion and regeneration was well investigated. For example, RTN4 has been shown to be involved in inhibition of the neurite outgrowth in human cell lines (*Fournier et al., 2001*) and inhibition of RTN4A via an antibody (ab) can improve visual recovery after retinal injury in mice (*Baya Mdzomba et al., 2020*). RTN1, in contrast to RTN4, did not inhibit axonal regeneration (*GrandPré et al., 2000*), which correlates with our previous observations that RTN1A might be involved in axonal elongation of cutaneous nerves in prenatal mouse skin (*Cichoń et al., 2020*). In humans, a possible interaction between the cytoskeleton and RTN4 in monocyte-derived Mφs for instance has been pointed out (*Schanda et al., 2011*). Based on previous findings, we aimed to add understanding on the function of RTN1A in the ER of LCs and study its involvement in structural remodeling. To identify the function and potential involvement of RTN1A-induced morphological changes, we studied its behavior in immune responses to specific TLR agonists on/in resident LCs (rLCs) and emigrated LCs (eLCs) in human epidermal explants.

## Results
### RTN1A is involved in the dynamic of dendrite retraction in LCs
In the first set of experiments we studied morphological changes in rLCs upon hampering RTN1A functionality with an α-RTN1A ab in human epidermal sheets ex vivo (*Figure 1A*). Incubation of dermatomed skin with the enzyme dispase II dissociated the basement membrane, thus enabling separation of the epidermis from the dermis and consequently exposure of epidermal cells to abs. The untagged α-RTN1A ab was detected with a fluorescently labeled secondary ab in the cell body

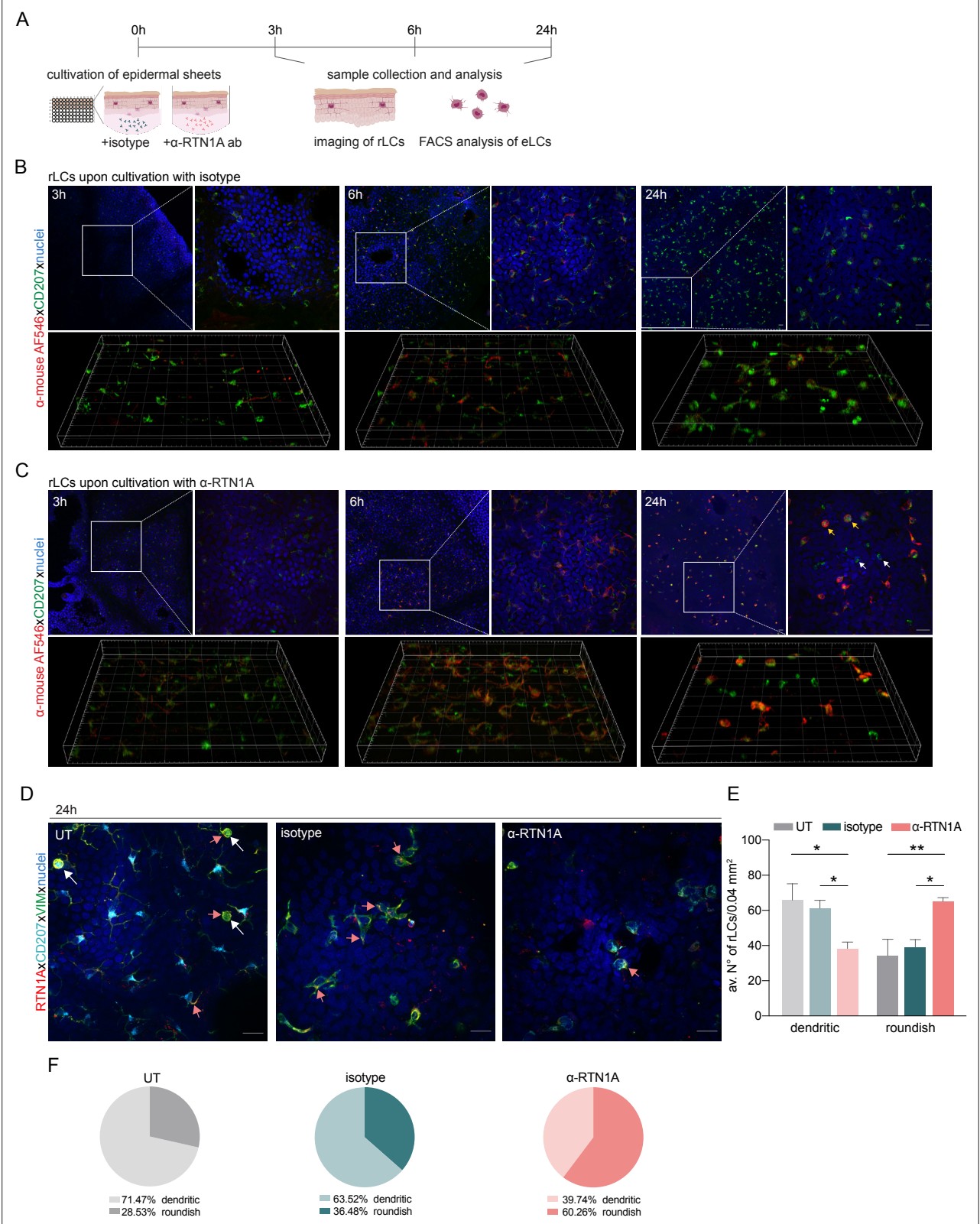

**Figure 1.** Impairment of Reticulon 1A (RTN1A) functionality instigates a roundish morphology in resident Langerhans cells (rLCs). (**A**) Experimental workflow demonstrating the cultivation of human epidermal sheets with an α-RTN1A antibody (ab) and the isotype for indicated time points and subsequent analysis strategies. (**B–C**) Representative immunofluorescence (IF) images of isotype- and α-RTN1A-treated epidermal sheets at indicated time points stained for CD207 (green), a secondary ab (red) to visualize the uptake of the isotype and α-RTN1A ab by rLCs and nuclear counterstaining

*Figure 1 continued on next page*

*Figure 1 continued*

with 4',6-diamidino-2-phenylindole (DAPI) (blue). Zoom-ins of the boxed areas are also shown as 3D projections underneath. RTN1A⁺ rLCs: yellow arrows, RTN1A⁻ rLCs: white arrows. n=3, scale bar: 20 μm. (**D**) Representative IF images showing RTN1A, CD207, vimentin, and DAPI staining in untreated (UT), isotype- and α-RTN1A-treated human epidermal sheets after 24 hr of cultivation. Co-localization of RTN1A with vimentin: pinkish arrows, rLCs with partially retracted dendrites: white arrows. n=4, scale bar: 20 μm. (**E, F**) Enumeration, percentage, and distribution of dendritic and roundish rLCs in epidermal sheets upon 24 hr of culture and indicated treatment. (**E**) Data are shown as standard error of the mean (SEM) from four fields of view (FOVs) of four donors and were analyzed using two-way ANOVA with Tukey's multiple-comparison test. (**F**) Data represent mean of four donors. *p≤0.05, **p≤0.01.

The online version of this article includes the following figure supplement(s) for figure 1:

**Figure supplement 1.** The frequency of RTN1A protein expression in resident Langerhans cells (rLCs) and emigrated LCs (eLCs).

and in the dendrites of rLCs already after only 3 hr, more pronounced at 6 hr and most prominent in the majority of rLCs at 24 hr of cultivation, while the isotype signal was substantially weaker (6 hr) or undetectable (3 and 24 hr) (*Figure 1B and C*). A prominent roundish morphology of rLCs was observed after 24 hr of incubation with the α-RTN1A abs in comparison to isotype-treated epidermis. rLCs captured the α-RTN1A ab (*Figure 1C*, yellow arrows). Notably, not all rLCs did so (*Figure 1C*, white arrows). Indeed, not all rLCs and eLCs express RTN1A to which the ab could have bound during the culture period. The frequency of RTN1A expression was determined in freshly isolated rLCs and eLCs by flow cytometry (*Figure 1—figure supplement 1A*). Around 80% of CD1a⁺CD207⁺ rLCs and eLCs express RTN1A (*Figure 1—figure supplement 1A*). Gating of RTN1A⁺ cells according to LC marker expression revealed that ~70% and ~80% of rLCs co-express CD207 and CD1a, respectively (*Figure 1—figure supplement 1B*). A similar observation was found for eLCs (~70% of RTN1A⁺ eLCs co-express CD207 or CD1a; *Figure 1—figure supplement 1C*).

For further analysis of rLCs in this experimental setup, we employed markers such as the C-type lectin receptor CD207 (can be localized in the plasma membrane and intracellularly) (*Hunger et al., 2004*), RTN1A (distributed in the ER) and vimentin to label intermediate filaments (major component of the cytoskeleton) (*Mahrle et al., 1983*). Images of untreated (UT) epidermal sheets cultured for 24 hr showed co-localization of RTN1A with vimentin in the cell body and in the most distant tips of

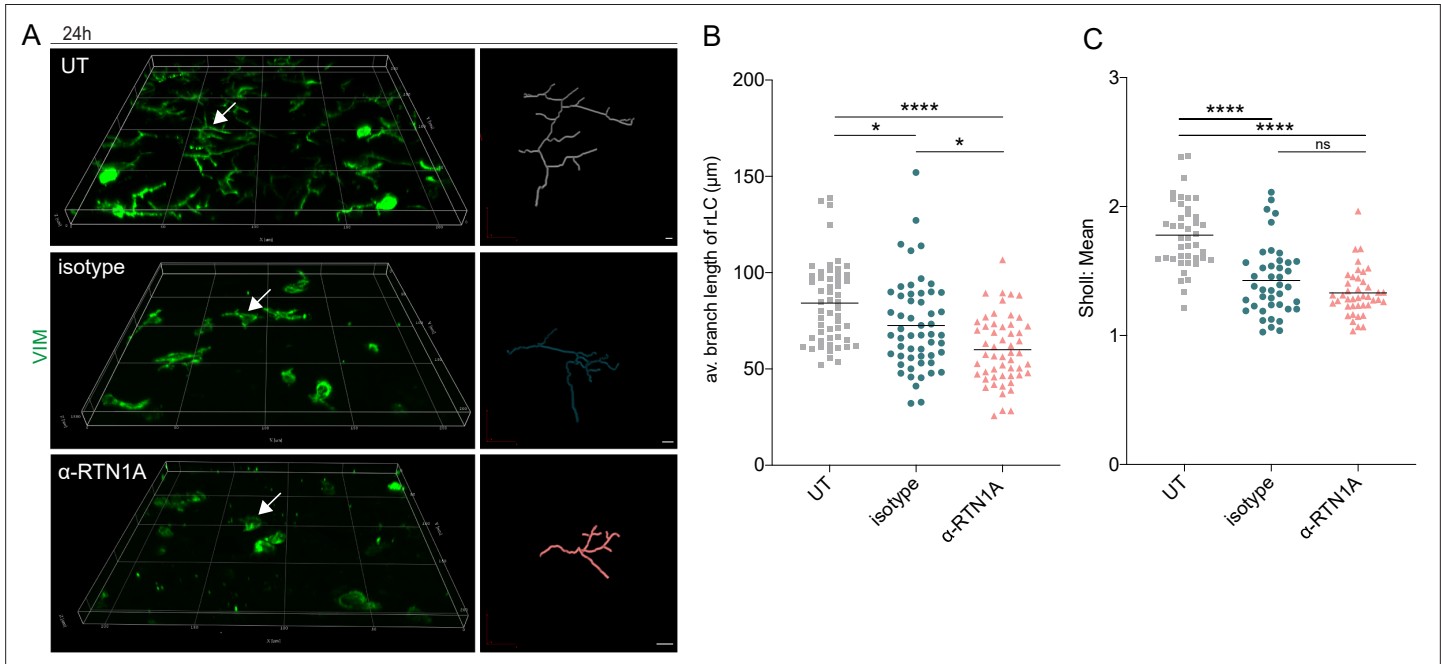

**Figure 2.** Inhibition of Reticulon 1A (RTN1A) in resident Langerhans cells (rLCs) significantly alters dendrite length and distribution. (**A**) 3D projections of epidermal sheets upon 24 hr of culture, indicated treatment, and vimentin (VIM) staining (left panel). Shown are single rLC 3D trajectories based on intermediate filament expression (right panel). Scale bar: 10 μm. (**B, C**) Evaluation of the average rLC branch lengths and Sholl analysis of rLC dendricity upon 24 hr of culture and indicated treatments. Four fields of view (FOVs) were evaluated of four donors and data analyzed using two-way ANOVA with Tukey's multiple-comparison test, ns = not significant. *p≤0.05, ****p≤0.0001.

dendrites with dot-like RTN1A accumulations (*Figure 1D*, pinkish arrows), whereas CD207 expression was detected at lower intensity and mainly the cell body (*Figure 1D*). We also captured cells with halfway-retracted dendrites most likely representing migratory LCs (*Figure 1D*, white arrows). Upon 24 hr of incubation with α-RTN1A abs, rLCs revealed a consistent and recurrent reduction of dendricity in comparison to isotype-treated and UT epidermal sheets (*Figure 1D*). Enumeration of dendritic and roundish rLCs within epidermal biopsy punches after 24 hr of incubation with or without abs showed a significant decrease in the number (*Figure 1E*) and percentage (UT: 71.47; isotype: 63.52; α-RTN1A ab: 39.74; *Figure 1F*) of dendritic rLCs after blocking of RTN1A in comparison to controls. This observation correlates with a significant increase of roundish rLCs (UT: 28.53; isotype: 36.48; α-RTN1A ab: 60.26; *Figure 1F*).

Next, we compared morphological changes in rLCs in UT, isotype, and α-RTN1A ab-treated epidermal sheets. Using 3D trajectories of intermediate filaments (vimentin) in rLCs (*Figure 2A*), we analyzed cell dendricity and the frequency of dendrite distribution in rLCs (*Figure 2B and C*). Quantification of the dendrite lengths in 3D fluorescent images of cultures epidermal sheets revealed that inhibition of RTN1A caused significant dendrite retraction in rLCs compared with those in isotype and UT epidermal sheets (*Figure 2B*). To address also the frequency and complexity changes of rLC

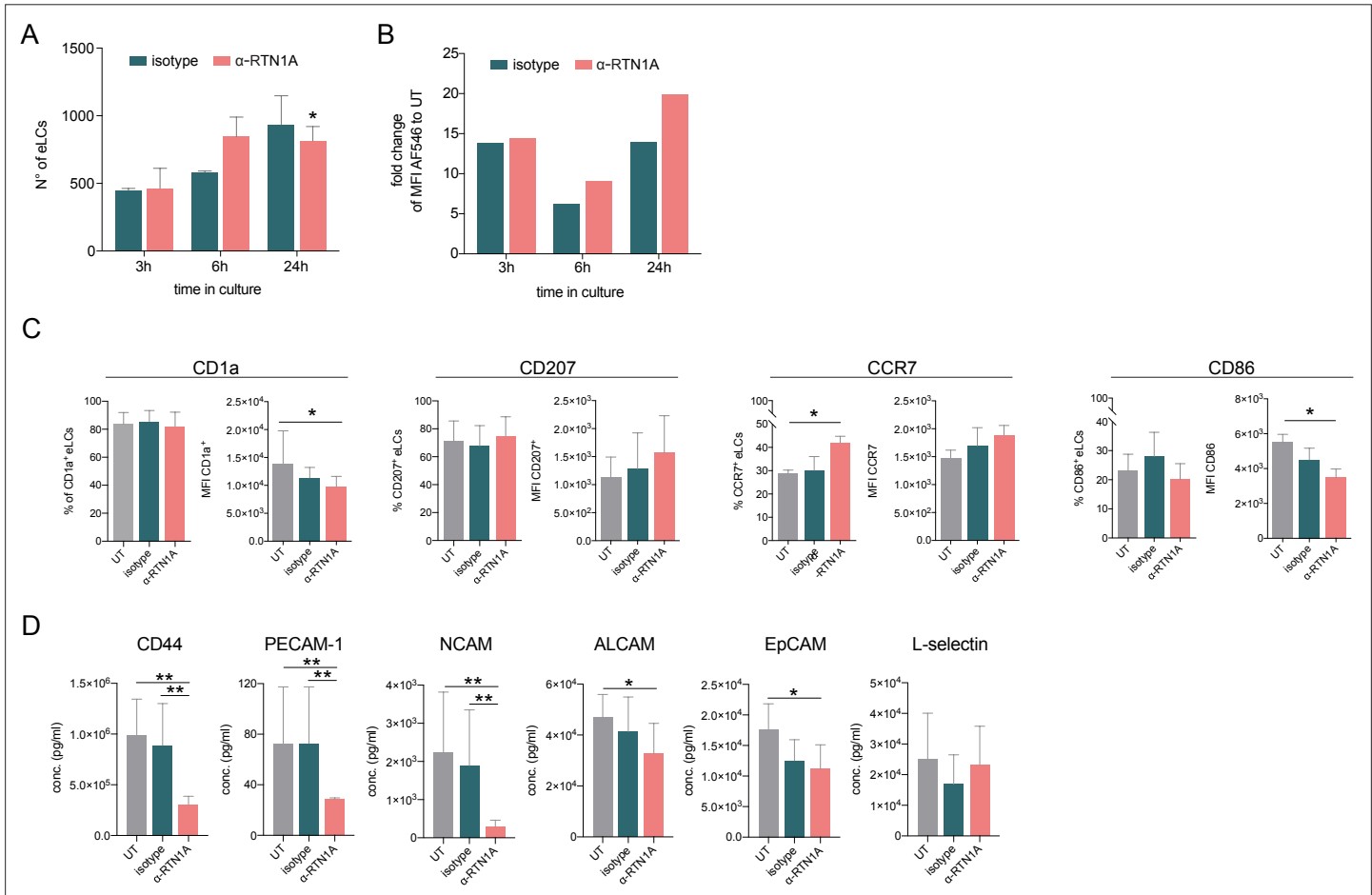

**Figure 3.** Hampering Reticulon 1A (RTN1A) function decreases the Langerhans cell (LC) migration potential and alters the marker expression in emigrated LCs (eLCs). (**A, B**) Enumeration and ab uptake of pre-gated CD207⁺CD1a⁺ eLCs, collected from culture wells with epidermal sheets at indicated time points and analyzed via flow cytometry. (**A**) Data are presented as standard error of the mean (SEM) from triplicates of three donors. Ordinary one-way ANOVA with Tukey's multi-comparison test was used. *p≤0.05. (**B**) Data from triplicates, including three donors, are shown as mean fold change to untreated (UT) eLCs. (**C**) Marker expression profile of pre-gated CD207⁺ eLCs upon 24 hr of culture, treatment, and flow cytometry analysis. Data shown represent mean ± SEM of three donors and were analyzed using two-way ANOVA Tukey's multi-comparison test. *p≤0.05. (**D**) Adhesion molecule concentrations in supernatants of cultivated epidermal sheets with indicated 24 hr treatments and subsequent LEGENDplex bead array measurement. Data are shown as SEM from duplicates of four donors and were analyzed using two-way ANOVA with Tukey's multiple-comparison test. *p≤0.05, **p≤0.01.

dendrites, we further employed the Sholl analysis which quantifies dendritic arbors is series of spheres from the cell body toward the dendrites (*Sholl, 1953*). Application of this analysis for rLCs in epidermal sheets showed significantly decreased rLC dendrite distribution in isotype- and α-RTN1A ab-treated epidermal sheets in comparison to UT epidermal sheets (*Figure 2C*). Of note, the difference between isotype- and α-RTN1A ab-treated epidermal sheets was not significant, indicating that the inhibition of RTN1A affected the length of rLC dendrites but not the frequency and complexity (*Figure 2C*).

## Despite dendrite retraction, rLCs with blocked RTN1A function remain in the epidermis

To evaluate whether blocking of RTN1A with the inhibitory ab affects LC emigration from epidermal sheets, we analyzed the number of eLCs at selected time points. We found a significant decrease in the migration potential of LCs in culture wells containing α-RTN1A ab-treated compared to isotype-treated epidermal sheets after 24 hr of cultivation (*Figure 3A*). Similar to rLCs (*Figure 1B and C*), ab uptake was also detected in eLCs at all investigated time points, with a tendency of a more pronounced though not significantly higher signal of RTN1A ab compared to the isotype (*Figure 3B*). We assume that LCs carried the α-RTN1A/isotype abs after detachment from the epidermis. However, it is also conceivable that some eLCs have taken up abs in the medium after the emigration (*Figure 3B*). Next, we analyzed typical LC markers in eLCs (*Figure 3C*) such as CD1a, a microbial lipid-presenting molecule (*Kim et al., 2016*; *Van Rhijn et al., 2015*; *Hunger et al., 2004*), as well as CD207. We found that the CD1a expression level (mean fluorescence intensity [MFI]), but not the percentage of CD1a+ eLCs, was significantly reduced after α-RTN1A ab treatment, whereas CD207 expression was unchanged. The migratory and mature phenotype of LCs is distinguishable by increased expression levels of CCR7 and co-stimulatory molecules such as CD86 (*Förster et al., 2008*). Here, the inhibition of RTN1A caused a significant increase in the percentage and expression intensity of CCR7, and a decrease of CD86 in comparison to controls. These results suggest that inhibition of RTN1A endorse acquisition of the migratory phenotype by LCs, along with preventing the maturation of some LCs. To better understand the mechanism involved in rLC morphology changes after blocking RTN1A functionality and LC migration, we comparatively measured the presence of key adhesion molecules in culture supernatants after 24 hr. The adhesion molecule CD44 can stimulate intracellular calcium mobilization, and actin- (*Wang et al., 1999*) and vimentin-mediated cytoskeleton remodeling (*Päll et al., 2011*). PECAM-1 (*Hu et al., 2016*) plays a role in endothelial cell-cell adhesion (*Dasgupta et al., 2009*) and NCAM promotes neuron-neuron adhesion and neurite outgrowth (*Frese et al., 2017*). ALCAM facilitates attachment of DCs to endothelial cells (*Iolyeva et al., 2013*) and migration of other endothelial cells (*Ikeda and Quertermous, 2004*), EpCAM regulates LC adhesion and foster their migration (*Gaiser et al., 2012*; *Eisenwort et al., 2011*). L-selectin is a calcium-dependent lectin expressed by leukocytes and mediates cell adhesion by binding to neighboring cells (*Bernimoulin et al., 2003*; *Wedepohl et al., 2017*). Treatment of epidermal sheets with the α-RTN1A ab for 24 hr significantly decreased the concentrations of CD44, PECAM-1, NCAM, ALCAM, and EpCAM but not L-selectin in culture supernatants in comparison with controls (*Figure 3D*). This assay cannot be used to assign a selected molecule to a particular cell type. These data suggest that the detachment of rLCs from the tissue was hampered.

## Expression of RTN1A substantially changes the cell size of myeloid cells

Next, we assessed the involvement of RTN1A in cytoskeletal remodeling by determining shape and morphology changes upon expression of human RTN1A in the RTN1A⁻ monocyte-like cell line THP-1 (*Figure 4A*). After confirming protein expression by flow cytometry, THP-1 RTN1A+ cells and THP-1 wild-type (wt) cells were used for further experiments (*Figure 4A*). THP-1 RTN1A+ cell morphology was considerably altered compared to THP-1 wt cells as they were significantly smaller in size and had markedly more condensed intermediate filament structures as visualized by vimentin staining (*Figure 4B, C* and *Figure 4—figure supplement 1A*). To assess a potential cross-talk between RTN1A and the cytoskeleton, THP-1 RTN1A+ cells were cultivated on fibronectin and imaged to analyze co-localization of RTN1A with vimentin and F-actin in three-cell compartments (bottom, middle, and top) (*Figure 4D*; *Figure 4—figure supplement 1B*). We found a significant overlap between RTN1A and vimentin at the bottom and top of the cell (*Figure 4D, E*), whereas there was less co-localization with

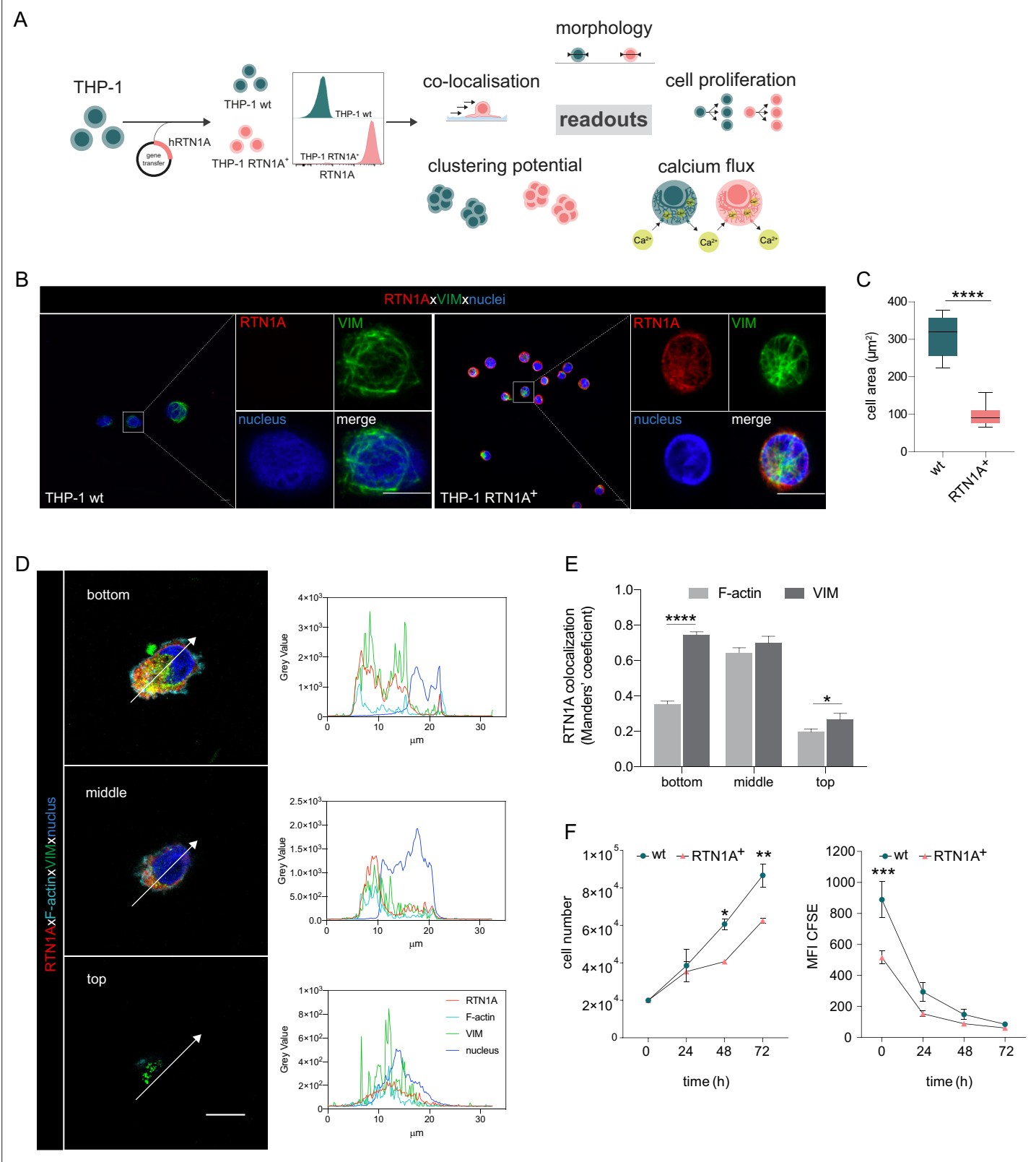

**Figure 4.** Expression of Reticulon 1A (RTN1A) in the myeloid THP-1 cell line affects the cell size. (**A**) Workflow for the generation of THP-1 RTN1A[+] cells and their comparative analysis with THP-1 wild-type (wt) cells. (**B**) Representative immunofluorescence (IF) images of THP-1 wt and THP-1 RTN1A[+] cells on adhesion slides stained for RTN1A, vimentin (VIM), and nuclei (4',6-diamidino-2-phenylindole [DAPI]). n=4; scale bar: 10 μm. (**C**) Comparative evaluation of the cell area revealed substantial divergences between THP-1 wt and THP-1 RTN1A[+] cells. Data are shown as standard error of the mean

*Figure 4 continued on next page*

*Figure 4 continued*

(SEM) from four fields of view (FOVs; n=4) and were analyzed using unpaired, two-tailed Student's t test. ****p≤0.0001. (D, E) Representative IF images and quantification using Manders' coefficient of RTN1A co-localization with filamentous proteins in a THP-1 RTN1A⁺ cell within three-cell compartments: bottom, middle, and top of the cell (right panel). Scale bar: 10 μm. Data are shown as SEM (10 cells/4 FOVs; n=2), and analyzed using two-way ANOVA with Tukey's multiple-comparison test. *p≤0.05, ****p≤0.0001. (F) Evaluation of the cell number and proliferation rate of THP-1 wt and THP-1 RTN1A⁺ cells within the time period indicated. Data presented as SEM were analyzed with two-way ANOVA, Sidak's multiple-comparison test (n=3). *p<0.05, **p≤0.01, ***p≤0.001.

The online version of this article includes the following figure supplement(s) for figure 4:

**Figure supplement 1.** Expression of RTN1A in THP-1 cells leads to constriction of intermediate filaments and their colocalization with RTN1A.

F-actin in the same cell compartments (*Figure 4D, E*). In the middle part of the cell, RTN1A showed similar co-localization levels with F-actin and vimentin (*Figure 4D, E*). As during cultivation THP-1 wt and THP-1 RTN1A⁺ cells showed different growth dynamics, we subsequently evaluated their proliferation rate and cell growth. Indeed, THP-1 RTN1A⁺ cells displayed a significantly lower proliferation rate, CFSE dye dilution, and cell number compared with THP-1 wt control (*Figure 4F*). The involvement of RTN1A in determination of morphological features such as cell size and dendricity was further studied in differentiated adherent THP-1 RTN1A⁻ wt Mφs and THP-1 RTN1A⁺ Mφs (*Figure 5A, B*). Expression of RTN1A in THP-1 Mφs resulted in altered morphology with significantly larger cell bodies and substantially longer cell protrusions when compared with THP-1 wt Mφs (*Figure 5B–D*). Alike in undifferentiated THP-1 cells (*Figure 4D, E*), RTN1A significantly co-localized with vimentin and to a lesser extent with F-actin (*Figure 5E*).

## RTN1A inhibits calcium flux and regulates cell adhesion in myeloid cells

We discovered that THP-1 RTN1A⁺ cells display an increased capacity in cellular aggregate formation compared with THP-1 wt cells (*Figure 5F and G* and *Figure 5—figure supplement 1*). In line with this observation, we analyzed cell culture supernatants for the presence of adhesion molecules. We found increased NCAM and PECAM-1 and significantly decreased ALCAM concentrations in supernatants of THP-1 RTN1A⁺ in comparison to THP-1 wt cells (*Figure 5H*). The ER is an important calcium ion store and proper calcium levels are crucial for maintaining balanced cell functions (*Vandecaetsbeek et al., 2011*; *Shibaki et al., 1995*). Calcium ions (Ca²⁺) are a versatile second messenger involved in signal transduction and controlling activity of adhesion molecules, such as CD44, L-selectin (*Ivetic et al., 2019*; *Berridge et al., 2003*). To analyze whether an altered calcium homeostasis in THP-1 RTN1A⁺ cells promotes the enhanced aggregation capacity, we comparatively monitored Ca²⁺ flux in THP-1 wt and THP-1 RTN1A⁺ cells using ratiomeric calcium flux measurement (*Wendt et al., 2015*) with the Ca²⁺ indicator Fura-3 red. Calcium ionophores such as ionomycin are binding calcium ions (*Liu, 1978*), which induces opening calcium stores and reaugmenting of [Ca⁺]ᵢ (*Kao et al., 2010*). Both the early and late response to ionomycin in THP-1 RTN1A⁺ cells was significantly lower in comparison to the calcium flux in THP-1 wt cells (*Figure 5I*). Next, we tested whether RTN1A plays a role in differently induced Ca²⁺ mobilization from the ER and therefore applied thapsigargin (*Liu, 1978*). This compound inhibits sarcoplasmic reticulum/ER Ca²⁺-ATPase (SERCA) channels, resulting in a clearly distinguishable depletion of intracellular Ca²⁺ stores in the early phase and subsequent activation and opening of plasma membrane calcium channels in the late phase. Cell stimulation with thapsigargin caused significantly lower Ca²⁺ efflux from the ER store in the early and late phase of the measurement in THP-1 RTN1A⁺ cells (*Figure 5J*). The results show that RTN1A has an inhibitory effect on the Ca²⁺ efflux from ER in comparison to THP-1 wt cells and most likely results in a decreased Ca²⁺ influx through the plasma membrane. Together, our data suggest that RTN1A expression may induce a feedback regulatory mechanism to counteract elevated cell adhesion and further highlights a potential role of RTN1A in fine-tuning cell activation and adhesion. These results comply with a previous finding of the inhibitory effect of RTN1A on calcium release and therefor calcium flux in nerve cells (*Kaya et al., 2013*) and suggest its involvement in myeloid cell adhesion via regulation of integrin activation (*Sjaastad et al., 1996*).

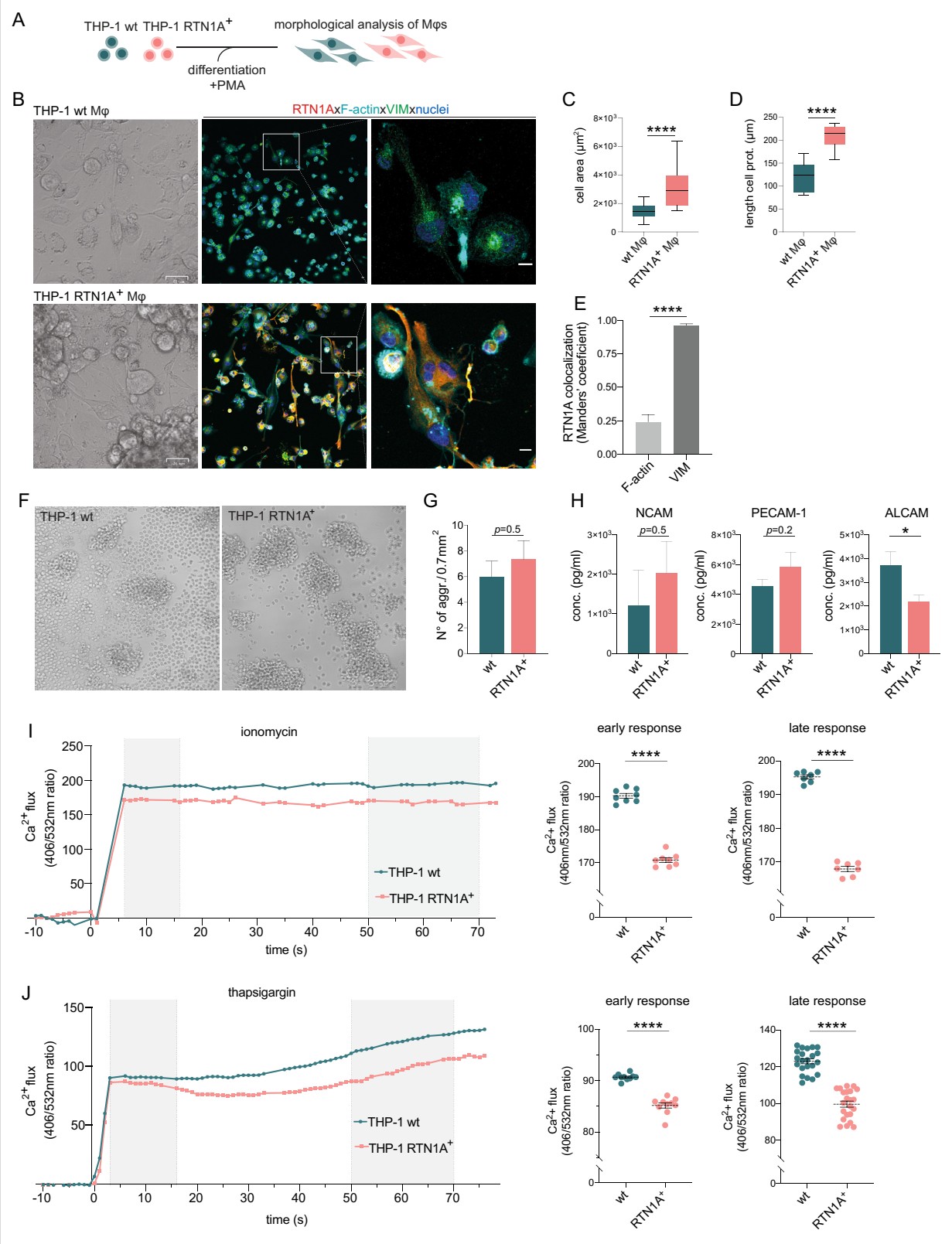

**Figure 5.** Reticulon 1A (RTN1A) considerably alters the morphology of THP-1 RTN1A+ macrophages (Mφs) as well as enhances aggregate formation and impairs Ca²⁺ flux in THP-1 RTN1A+ cells. (**A**) Workflow for the differentiation and comparative analysis of THP-1 RTN1A+ Mφs and THP-1 wild-type (wt) Mφs. (**B**) Representative bright-field (BF) and immunofluorescence images (IF; co-staining with RTN1A, F-actin, vimentin [VIM], and 4',6-diamidino-2-phenylindole [DAPI]) of THP-1 wt Mφs and THP-1 RTN1A+ Mφs. Scale bar: BF: 31 and IF: 10 µm, n=4. (**C, D**) Comparative analysis of cell body size and

*Figure 5 continued on next page*

Figure 5 continued

average length of cell protrusions of differentiated THP-1 wt Mφs and THP-1 RTN1A⁺ Mφs. Data were analyzed using unpaired, two-tailed Student's t test, n=4. ****p≤0.0001. (E) Comparative co-localization between RTN1A, F-actin, and VIM using Manders' coefficient. Data shown are standard error of the mean (SEM), n=4. ****p≤0.0001. (F) Representative BF image of THP-1 wt and THP-1 RTN1A⁺ cells forming aggregates during culture. Scale bar: 100 µm. (G) Enumeration of THP-1 wt and THP-1 RTN1A⁺ cell aggregates. Data are demonstrated as SEM from four fields of view per passage and analyzed using unpaired, two-tailed Student's t test, n=6. (H) Adhesion molecule concentrations in supernatants after 48 hr of THP-1 wt and THP-1 RTN1A⁺ cell cultivation, measured in duplicates with LEGENDplex bead array. Data are shown as SEM from three different passages and were analyzed using unpaired, two-tailed Student's t test. *p≤0.05. (I, J) Ratiomeric analysis of early and late phase calcium flux in THP-1 wt and THP-1 RTN1A⁺ cells using Fura-3 calcium indicator, ionomycin (n=5), and thapsigargin (n=4). Unpaired two-tailed t test was used for the response to both ionomycin and thapsigargin. ****p≤0.0001.

The online version of this article includes the following figure supplement(s) for figure 5:

**Figure supplement 1.** Enumeration of small and big THP-1 wild-type (wt) and THP-1 Reticulon 1A⁺ (RTN1A⁺) cell aggregates per 0.7 mm² from four fields of view (FOVs).

## Stimulation of TLR1/2, TLR2, and TLR7 significantly diminishes RTN1A expression levels in eLCs and induce rLC clustering

We next explored whether a relation exists between RTN1A and the activation of LCs via TLR agonists, thus mimicking inflammatory conditions in ex vivo human skin. This was of great interest as it may reflect the everyday life of human skin which is constantly exposed to myriad environmental assailants. Incubation of epidermal explants with selected extra- and intracellular TLR agonists and subsequent analysis of eLCs by flow cytometry (*Figure 6A* and *Figure 6—figure supplement 1A*) revealed that both the percentage of RTN1A⁺ eLCs and RTN1A expression intensity in eLCs were significantly diminished after activation with the agonists TLR1/2 (Pam3CSK4), TLR2 (*Listeria monocytogenes*), and TLR7 (imiquimod, polyadenylic:polyuridylic acid (poly(A:U)), but not to agonists of TLR2/6 [mycoplasma salivarium] and TLR3 [low and high molecular weight polyinosinic:polycytidylic acid, LMW and HMW poly(I:C)]) (*Figure 6B–D*). Of note, the downregulation of RTN1A in eLCs was transient, since we observed a tendency for recovery of RTN1A protein expression upon 48 hr of cultivation (*Figure 6E*). Next, we examined whether eLCs diminish RTN1A expression during the activation process. In cultures with UT epidermal sheets, we observed a small percentage of activated eLCs (*Figure 6—figure supplement 1B, C*). This is in line with previous observations that some rLCs can be activated after enzymatic separation and cultivation (*Pearton et al., 2010*). Epidermal sheets cultured with TLR1/2, TLR2/6, and TLR3/7 agonists and analysis of pre-gated RTN1A⁺ eLCs revealed an upregulation of CD83 and CD86 in comparison to the UT control (*Figure 6—figure supplement 1B, C*). Of note, TLR1/2 stimulation significantly enhanced the percentage of activated CD83⁺CD86⁺ eLCs (*Figure 6—figure supplement 1B, C*). These data provide evidence that the decreased RTN1A expression correlates with the activation status of LCs and implies an active communication between TLRs and RTN1A. Next, the expression and distribution of RTN1A, CD86, and CD83 were examined in/on rLCs in epidermal sheets after cultivation with selected extra- and intracellular TLR agonists. Stimulation with TLR1/2, TLR2, and TLR7 agonists induced the formation of rLC clusters and were repeatedly detected in the convex epidermal areas (*Figure 7A*, inserts). The big and small rLC clusters exhibited diminished dendrites and a roundish morphology (*Figure 7B*). Furthermore, we observed low co-localization between activation markers (cell membrane) and RTN1A (ER) in the single, activated, and dendritic rLCs (*Figure 7B*). In contrast, upon stimulation of epidermal sheets with TLR2/6 and TLR3 (LMW and HMW p(I:C)) agonists, the morphology of rLCs was unaltered and comparable to the UT control. The expression intensity of RTN1A in rLCs was slightly but not significantly reduced upon stimulation with all TLR agonists in comparison to the UT control (*Figure 7C*). Significant upregulation of CD83/CD86 was observed after TLR7 stimulation (*Figure 7D*). Further analysis of clusters formed by rLCs upon TLR1/2 and TLR7 stimulation of epidermal sheets revealed that they were not proliferating (*Figure 7—figure supplement 1A*). They simultaneously acquired a migratory phenotype and upregulated either MMP-9 (*Ratzinger et al., 2002*) or CCR7 or co-expressed both markers (*Figure 7E*, yellow arrow: MMP-9/CCR7, red arrow: MMP-9, and turquoise arrow: CCR7). Stimulation of TLRs activates signaling cascades inducing proinflammatory response such as secretion of cyto- and chemokines (*Wang et al., 1999*). Accordingly, we have assessed epidermal sheet culture supernatants and detected low IL-6 levels with exception of a significant increase after stimulation with TLR3/7 poly(A:U). IL-8 and TNF-α concentrations were significantly elevated after stimulation with

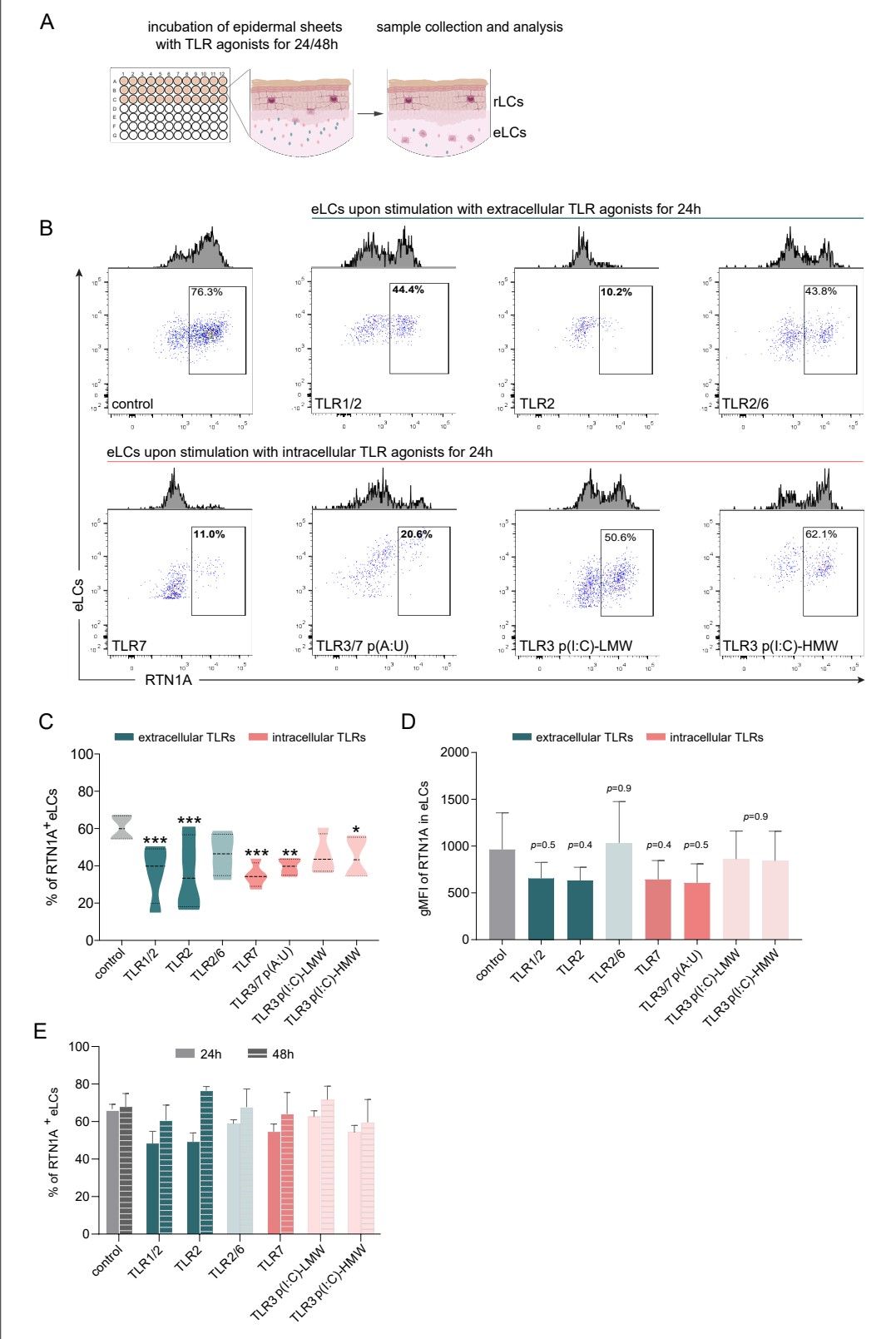

**Figure 6.** Emigrated LCs (eLCs) significantly decrease Reticulon 1A (RTN1A) expression upon stimulation with TLR1/2, TLR2, and TLR7 agonists. (**A**) Graphical presentation for the stimulation of epidermal sheets with TLR agonists and analysis. (**B**) Representative FACS blots of RTN1A expression in pre-gated eLCs upon stimulation of epidermal sheets with TLR1/2 (Pam3CSK4), TLR2 (*Listeria monocytogenes*), TLR2/6 (mycoplasma salivarium),

*Figure 6 continued on next page*

*Figure 6 continued*

TLR7 (imiquimod), TLR3/7 [polyadenylic:polyuridylic acid (poly(A:U))], and TLR3 [low and high molecular weight polyinosinic:polycytidylic acid (LMW and HMW poly(I:C))] for 24 hr. (**C, D**) The percentage and geometric mean fluorescence intensity (gMFI) of RTN1A in eLCs are shown as standard error of the mean (SEM) of triplicates, and analyzed using two-way ANOVA with Tukey's multiple-comparison test. *p≤0.05, **p≤0.01, ***p≤0.001. Some p values were not significant (ns), yet indicative of a trend for the reduction of RTN1A expression intensity. (**E**) Recovery of the RTN1A protein expression (% of RTN1A⁺ eLCs) after 48 hr of cultivation with indicated TLR agonists. Data shown represent mean ± SEM of triplicates from three donors.

The online version of this article includes the following figure supplement(s) for figure 6:

**Figure supplement 1.** Expression pattern of RTN1A protein within TLR-activated eLCs and their quantification.

---

TLR3-HMW, and TLR3-LMW agonists, respectively. Of note, MCP-1 levels were significantly reduced after stimulation with TLR2 and TLR3/7 poly(A:U) agonists in comparison to UT control (*Figure 7F*). In contrast, IL-10, IL-23 concentrations were slightly but not significantly elevated in comparison to IL-6 and IL-8 (*Figure 7—figure supplement 1B, C*).

## Discussion

In this study, we investigated the involvement of RTN1A in structural remodeling and effects of immune responses against pathogens by stimulation of TLRs in rLCs and eLCs in human epidermis ex vivo. We showed that local attenuation of RTN1A functionality with an N-terminus-targeting ab caused significant changes in the morphology of rLCs, leading to a roundish cell body with reduced dendrite length and dendrite distribution. Labeling of rLCs in human skin ex vivo (*Tripp et al., 2021*) and in mouse skin in vivo (*Flacher et al., 2010*) has been successfully employed before, yet in this study we targeted an intracellular protein. These observations led us to conclude that RTN1A is promoting elongation of dendrites in rLCs. Similar results with involvement of RTN1 family members in dendrite formation have been reported for Purkinje cells in mice (*Shi et al., 2017*) and cutaneous nerves (*Cichoń et al., 2020*) in developing mouse skin. Of note, other RTN family member such as RTN4 isoforms are vice versa involved in inhibition of neurite regeneration and elongation as well as impairment in sphingomyelin processing (*Fournier et al., 2001*; *Chen et al., 2000*; *Baya Mdzomba et al., 2020*; *Pathak et al., 2018*). Our further observations on effects and consequences of RTN1A expression in THP-1 cells, such as significantly smaller cell size and a denser intermediate filament constellation, and in contrary in differentiated adherent THP-1 RTN1A⁺ Mφs the increased capacity to form long cell protrusions and cell body, support our hypothesis on the crucial role of RTN1A in cytoskeleton dynamics. Moreover, the excessive co-localization with vimentin (*Figure 1D*, *Figure 4A*, *Figure 5E*), but less with F-actin, could indicate an interaction between ER tubules and intermediate filaments, thereby facilitating changes in cell morphology, when RTN1A is inhibited or overexpressed. A previous observation has described that fascin, an actin bundling protein (F-actin), is crucial for the formation of dendritic processes in LCs. In line with their observation of cytoskeletal remodeling, our data unravel a broader complexity about the regulation of dendrites in LCs (*Ross et al., 1998*). Cytoskeletal components such as type III intermediate filaments (e.g. vimentin) (*Cooper, 2000*) were shown to (i) promote cell adhesion through governing integrin functions (*Ivaska et al., 2007*; *Mendez et al., 2010*), (ii) stabilize microtubule dynamics (*Schaedel et al., 2021*), and (iii) endorse cell migration (*Messica et al., 2017*) by contact-dependent cell stiffening (*Battaglia et al., 2018*). Alterations in the adhesion molecule profile after inhibition of RTN1A in rLCs and THP-1 RTN1A⁺ cells further support our assumption that RTN1A is crucial for structural cell dynamics. Based on our findings we suggest that RTN1A could interact with intermediate filaments by acting as GTPase adaptor molecule, which will be investigated in the future.

Our discovery that RTN1A in THP-1 cells significantly reduces Ca²⁺ flux after stimulation with ionomycin is in line with results showing that RTN1A impaired the Ca²⁺ flux in Purkinje cells by inhibition of the RYR calcium channel (*Kaya et al., 2013*) and SERCA2b (*van de Velde et al., 1994*). Apparently, RTN1A can inhibit calcium flux in cells with an extensive dendrite network such as Purkinje cells, THP-1 RTN1A⁺ cells (this study), and presumably LCs. Calcium signaling leads to cell activation (*Clapham, 2007*), and lower calcium flux is supposed to protect the cell from unnecessary activation and/or maintain their steady state within the tissue (*Marlin and Carter, 2014*; *Cichon et al., 2017*). Indeed, we

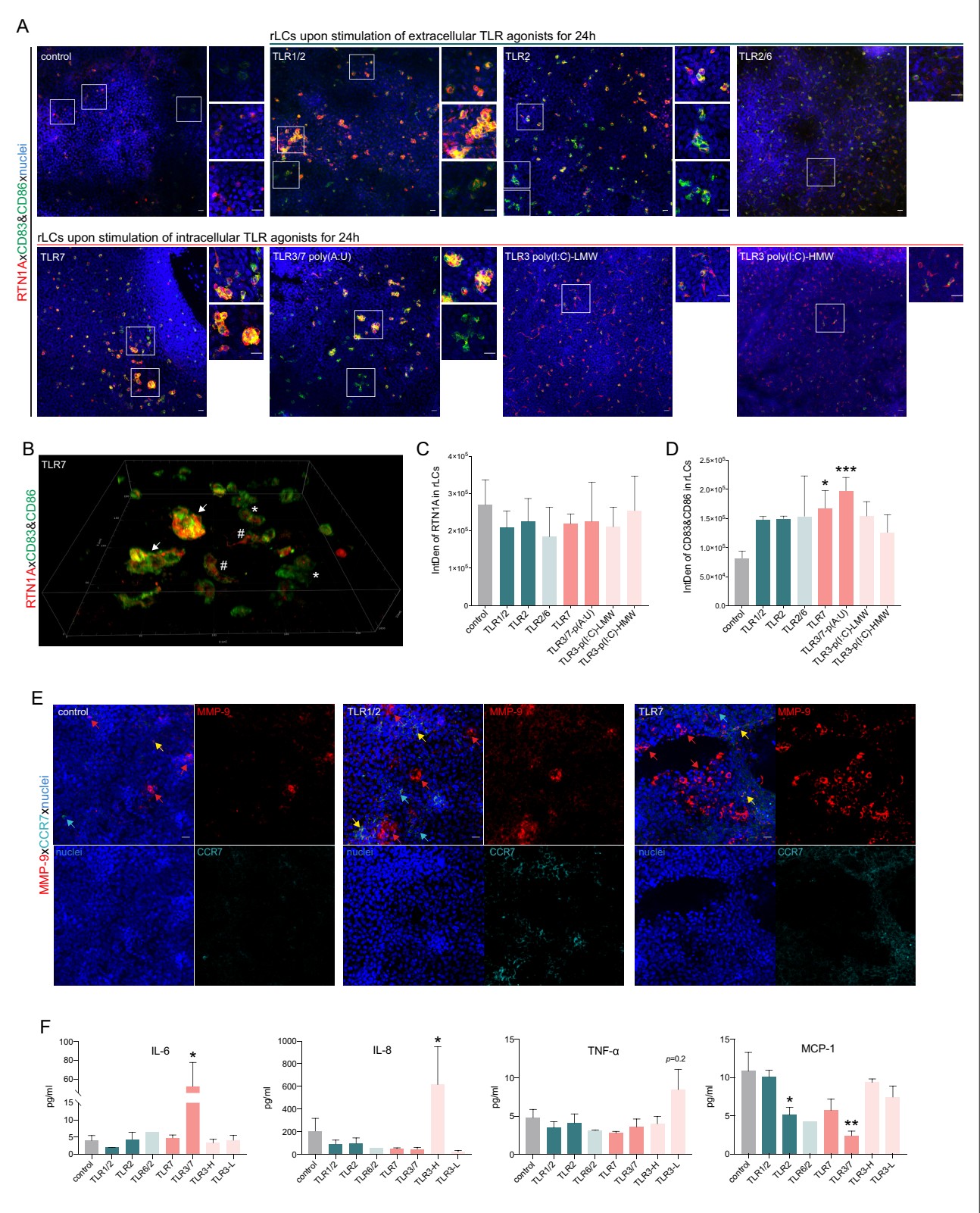

**Figure 7.** Stimulation of epidermal sheets with TLR1/2, TLR2, and TLR7 agonists initiates cluster formation of resident Langerhans cells (rLCs). (**A**) Representative immunofluorescence (IF) images of untreated (control) and indicated TLR-treated epidermal sheets upon 24 hr of culture stained with Reticulon 1A (RTN1A) (red), CD83, CD86 (green), and 4',6-diamidino-2-phenylindole (DAPI) (blue). n=5, scale bar: 20 μm. (**B**) 3D projection of rLCs in an epidermal sheet after incubation with a TLR7 agonist for 24 hr (**A**). rLCs form big (arrow) and small (asterisk) clusters yet were also visible as single

*Figure 7 continued on next page*

*Figure 7 continued*

activated dendritic rLCs (hashtag). (**C, D**) Expression intensity of indicated markers in rLCs upon culture and treatment. Data are shown as standard error of the mean (SEM) representing four fields of view of five donors, analyzed with two-way ANOVA Tukey's multiple-comparison test. *p≤0.05,***p≤0.001. (**E**) Representative IF images of untreated (control) and TLR-stimulated epidermal sheets stained for MMP-9, CCR-7, and 4′,6-diamidino-2-phenylindole (DAPI). n=2, scale bar: 20 µm. (**F**) Inflammatory cyto- and chemokine concentrations in supernatants of epidermal sheet cultures after 24 hr, measured in duplicates with LEGENDplex bead array. Data are shown as SEM of four donors and analyzed using two-way ANOVA with Durrett's multiple-comparison test. *p≤0.05, **p≤0.01.

The online version of this article includes the following figure supplement(s) for figure 7:

**Figure supplement 1.** Resident Langerhans cells (rLCs) do not proliferate in clusters upon TLR stimulation and produce elevated levels of IL-10 and IL-23.

found, that application of the inhibitory RTN1A ab to epidermal sheet cultures prevented an upregulation of CD86 and consequently activation of LCs.

Human LCs express a particular repertoire of extracellular TLRs such as TLR1/2, TLR2 (*Flacher et al., 2006*) for detection of bacterial lipids, and intracellular TLR7 (*van der Aar et al., 2007*) but not TLR3 (*Tajpara et al., 2018*) for recognition of viral RNA. While assessing the effects of extra- and intracellular TLR stimulation on the expression and distribution of RTN1A in rLCs in epidermal sheets, we found that stimulation with TLR1/2, TLR2, and TLR7 agonists but not with TLR2/6 and TLR3 agonists significantly diminished the percentage and expression intensity of RTN1A in eLCs in culture wells (*Figure 4C and D*). As activation of TLR1/2, TLR2, and TLR7 leads to the generation of endolysosoms (*Kim et al., 2016*; *Latz et al., 2004*), we postulate that the temporary downregulation of RTN1A in

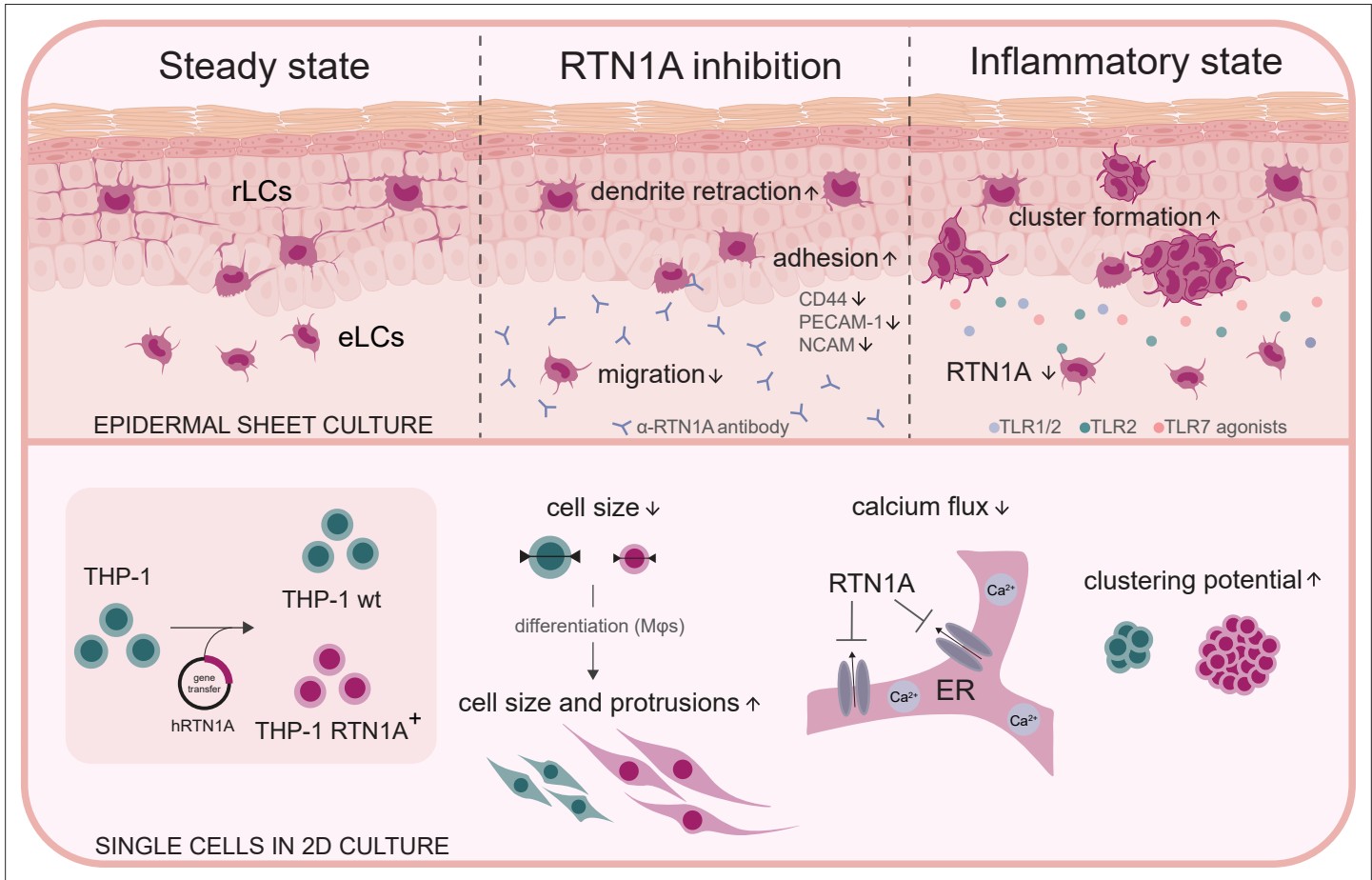

**Figure 8.** Summary and overview on the function of Reticulon 1A (RTN1A). Inhibition of RTN1A protein in resident Langerhans cells (LCs) in human epidermis ex vivo caused dendrite retraction, induced cell adhesion, and reduced LC migration. Stimulation of LCs with TLR agonists downregulated RTN1A expression and induced cluster formation (upper panel). Furthermore, expression of RTN1A in THP-1 cells altered cell size, morphology, calcium release from endoplasmic reticulum (ER) stores and cell aggregation in vitro (lower panel).

activated eLCs was due to the recruitment of RTN1A into endosomes. Essentially, it was shown that RTN3L (RTN3A), another member of the RTN family, is involved in endosome maturation (*Wu and Voeltz, 2021*) and in autophagy-induced fragmentation of ER tubules (*Grumati et al., 2017*). Furthermore, the morphological changes in activated eLCs could cause repression of the ER tubular network during the collision of late endosomes or lysosomes carried along microtubules (*Guo et al., 2018*), despite the fact that in unstimulated eLCs RTN1A expression is unchanged (*Figure 6A and B*). Moreover, stimulation of TLR1/2, TLR2, and TLR7, which impacts RTN1A expression, also induced cluster formation by rLCs within epidermal sheets (*Figure 5C*). The clustering rLCs acquired an activated and migratory phenotype (*Koch et al., 2006*; *Kobayashi et al., 1950*). To our knowledge, this behavior of rLCs in human epidermis has not been observed before. Previously, it was reported that dermal plasmacytoid DCs in mice can cluster in the dermis after topical stimulation with the TLR7 agonist imiquimod regulating antiviral response, but not epidermal LCs, which after several days of treatment appeared to be larger but not clustered (*Palamara et al., 2004*). A similar event was described for dermal CD11c[+] DCs but not for LCs in a contact dermatitis mouse model using hapten sensitization (*Natsuaki et al., 2014*). Recently, it was demonstrated that in the dermis of atopic dermatitis patients' perivascular leukocyte clusters can be infiltrated by other antigen-presenting cells to regulate T cell activation (*Peng et al., 2021*). As we have investigated the LC activation process in the early stages of inflammation-like conditions, minor levels of signature inflammatory cytokines produced usually by LCs and specific for the response to applied stimuli such as IL-1α, IL-1β, IL-6, IL-8, TNF-α (*Wang et al., 1999*), or IFN-γ (*Blasius and Beutler, 2010*) were measured.

In conclusion, as summarized in *Figure 8*, we demonstrated the importance of RTN1A in LC intra-tissue dynamics, suggesting its strong involvement in maintaining homeostasis in rLCs. Moreover, we demonstrate a close functional relation between RTN1A and particular TLRs at the active stage of LCs, which could have a protective role in the maintenance of tissue homeostasis.

# Materials and methods

**Key resources table**

| Reagent type (species) or resource | Designation | Source or reference | Identifiers | Additional information |
|---|---|---|---|---|
| Gene (*Homo sapiens*) | RTN1A | NM_0211369, Eurofins (this paper) | | |
| Cell line (*Homo sapiens*) | THP-1 | ATCC | TIB-202 | |
| Transfected construct (*Homo sapiens*) | pHR-SIN-BX-IRES-Emerald | *Paster et al., 2013* | | Lentiviral construct to transfect the THP-1 cell line and express RTN1A |
| Antibody | α-RTN1A (clone: mon162), unconjugated (Mouse monoclonal) | abcam | ab9274 | IF staining – 0.2 µg/ml; cultivation of epidermal sheets – 5 µg/ml/ |
| Antibody | α-RTN1A (clone: mon162), unconjugated (Mouse monoclonal) | Novus Biologicals, Biotechne | NBP1-97677 | IF staining – 0.2 µg/ml; cultivation of epidermal sheets – 5 µg/ml/ |
| Antibody | α-RTN1A-APC (clone: mon161, Mouse monoclonal) | Novus Biologicals, Biotechne | NBP1-97678AF647 | FACS analysis – 1:500 |
| Antibody | IgG1 [15-6E10A7] - Isotype control, unconjugated (Mouse monoclonal) | abcam | ab170190 | IF staining – 0.2 µg/ml; cultivation of epidermal sheets – 5 µg/ml/ |
| Antibody | α-CD207-FITC (clone 929F3.01, Rat monoclonal) | Dendritics | Cat:DDX0362 | IF staining – 0.25 µg/ml. 1:250 |

*Continued on next page*

*Continued*

| Reagent type (species) or resource | Designation | Source or reference | Identifiers | Additional information |
|---|---|---|---|---|
| Antibody | α-CD207, unconjugated (Rabbit monoclonal) | Sigma-Aldrich | HPA011216 | IF staining – 1:200 |
| Antibody | α-Human CD207-PE (clone DCGM4, Mouse) | Beckman Coulter | PN IM3577 | FACS analysis – 1:200 |
| Antibody | α-Human HLA-DR-PE (Mouse) | BD Pharmingen | Cat. 347401 | FACS analysis – 1:100 |
| Antibody | α-Human CD1a-PE (Mouse) | BD Pharmingen | Cat. 555807 | FACS analysis – 1:100 |
| Antibody | α-Human CD83-FITC (Mouse) | BD Pharmingen | Cat. 556910 | FACS analysis, IF staining – 1:100 |
| Antibody | α-Human CD86-FITC (Mouse) | BD Pharmingen | Cat. 555657 | FACS analysis, IF staining – 1:100 |
| Antibody | α-CCR7-APC | Miltenyi Biotech | 5171113456 | FACS analysis, IF staining – 1:100 |
| Antibody | Recombinant α-Vimentin-FITC (clone REA409) | Miltenyi Biotec | 130-116-508 | IF staining – 1:50 |
| Antibody | Recombinant FITC isotype control | Miltenyi Biotech | 130-113-449 | IF staining – 1:50 |
| Antibody | F(ab')2-Goat anti-Rabbit IgG (H+L) Cross-Adsorbed Secondary Antibody, Alexa Fluor 488 | Life Technologies | REF A-11070 | IF staining – 1:500 |
| Antibody | F(ab')2-Goat anti-Mouse IgG (H+L) Cross-Adsorbed Secondary Antibody, Alexa Fluor 546 | Invitrogen, Thermo Fisher Scientific | REF A-11018 | IF staining – 1:500 |
| Antibody | Alexa Flour 647F(ab′) 2 fragment of goat α-rabbit IgG (H+L) | Invitrogen, Thermo Fisher Scientific | REF A-21246 | IF staining – 1:500 |
| Commercial assay or kit | Human TLR kit | InvitroGen | Cat. tlrl-kit1hw | |
| Commercial assay or kit | LEGENDplex Human inflammatory Panel 1 (13-plex) w/VbP | Biolegend | Cat. 740809 | |
| Commercial assay or kit | LEGENDplex Human Adhesion Molecule Panel (13-plex) w/VbP | Biolegend | Cat. 740946 | |
| Chemical compound, drug | Thapsigargin | Invitrogen, Thermo Fisher Scientific | T7458 | |
| Software, algorithm | GraphPad Prism v8.0.1 | GraphPad Software | | |
| Software, algorithm | FlowJo v10.6.1 | BD (Becton, Dickinson & Company) | | |
| Software, algorithm | Fiji: ImageJ | *Schindelin et al., 2012* | | |
| Software, algorithm | ImarisViewer 9.8 | Oxford Instruments | | |
| Software, algorithm | Adobe Illustrator CS7 | Adobe Inc | | |
| Other | Alexa Fluor 647 Phalloidin | Invitrogen | REF A-22287 | IF staining – 1:500 |

*Continued on next page*

*Continued*

| Reagent type (species) or resource | Designation | Source or reference | Identifiers | Additional information |
|---|---|---|---|---|
| Other | Fixable Viability Dye eFluor 450 | eBioscience | Cat. 65-0863-18 | FACS analysis – 1:1000 |
| Other | DAPI (4′,6-diamidino-2-phenylindole dihydrochloride) | Sigma-Aldrich | Cat. D9542 | IF staining – 1:5000 |
| Other | Dispase II (neutral protease, grade II) | Roche Diagnostics | 04942078001 | 1.2 U/ml |
| Other | ProLong Gold antifade reagent | InvitroGen | Cat. P36934 | |
| Other | IC Fixation Buffer | eBioscience | Cat. 00-8222-49 | |
| Other | Permeabilization Buffer (×10) | eBioscience | Cat. 00-8333-56 | |
| Other | Phorbol 12-myristate 13-acetate (PMA), PKC activator | Abcam | Ab120297 | |
| Other | Triton X-100 | Sigma-Aldrich | T9284 | |
| Other | Fura Red, AM, cell permeant | Invitrogen, Thermo Fisher Scientific | F3020 | |
| Other | CellTrace CFSE Cell Proliferation Kit, for flow cytometry | Invitrogen, Thermo Fisher Scientific | C34554 | |
| Other | Ionomycin- Calcium ionophore – NFAT Activator | InvivoGen | inh-ion | |
| Other | Fibronectin Solution Human | PromoCell | Cat. C-43060 | |

## Processing of human skin

Experiments were performed within 1–2 hr after surgery. Skin was dermatomed (Aesculap) to 600 µm thickness, then incubated with 1.2 U/ml dispase II (Roche Diagnostics) in Roswell Park Memorial Institute 1640 medium (RPMI; Gibco, Thermo Fisher Scientific) with 1% penicillin/streptomycin (P/S; Gibco, Thermo Fisher Scientific) overnight (ON) at 4°C. After washing with phosphate buffered saline (PBS; Gibco, Thermo Fisher Scientific), epidermis was separated from the dermis. Epidermal punches with a diameter of 6 mm were obtained and used for the following experiments. The rLCs were isolated and purified as described previously (*Tajpara et al., 2018*).

## Cultivation of epidermal explants with TLR agonists and staining

Epidermal punch biopsies (6 mm diameter) in triplicates for each condition were floated on RPMI medium supplemented with 10% fetal bovine serum (FBS, Gibco), 1% P/S, and human TLR agonists (InvitroGen) in 96-well round bottom plates. The following TLR agonists were used: TLR1/2 (Pam3CSK4x3 HCl; 1 µg/ml), TLR2 (heat-killed *L. monocytogenes*; $10^8$ cells/ml), TLR3/7 ((poly(A:U)); 25 µg/ml), TLR3 (LMW and HMW poly(I:C); both at 0.8 µg/ml), TLR6/2 (mycoplasma salivarium, FSL-1, Pam2CGDPKHPKSF; 2.5 µg/ml) and TLR7 (imiquimod; 2.5 µg/ml). After 24 and 48 hr, epidermal sheets were collected, fixed with acetone (Merck) for 10 min at room temperature (RT), incubated with FITC-conjugated α-CD83 and α-CD86 abs, as well as with a primary mouse α-RTN1A ab (mon162, abcam) ON at 4°C. All abs were diluted in 2% bovine serum albumin (Gibco, Thermo Fisher Scientific) in PBS. Subsequently, sheets were incubated with an AF647-labeled α-mouse secondary ab (Thermo Fisher Scientific) for 1 hr at RT and counterstained with 4′,6-diamidino-2-phenylindole (DAPI; Sigma-Aldrich) for 1 min. After PBS washing, sheets were mounted (ProLong Gold antifade reagent, Invitrogen) with the *stratum corneum* facing the slide and imaged. At the same time points cultivation medium was harvested and stored at –80°C for further analyses and eLCs were processed and analyzed as described below.

## Cultivation of epidermal explants with an α-RTN1A ab

Epidermal punch biopsies (6 mm diameter) in triplicates were floated on medium containing either a mouse α-RTN1A ab (mon162, abcam) or the respective isotype control ab (IgG1, abcam) (5 µg/ml/each) in 96-well round bottom plates at 37°C, 5% $CO_2$. Sheets were collected at 3, 6, and 24 hr, fixed with acetone and stained with a secondary α-mouse cross-absorbed F(ab') ab fragment conjugated with AF546 (Thermo Fisher Scientific) and a FITC-labeled α-CD207 (Dendritics) to identify LCs. In some experiments, 24 hr cultured epidermal sheets were stained with a FITC-labeled α-vimentin, primary rat α-CD207 (Sigma-Aldrich) and mouse α-RTN1A abs (mon162, abcam) ON at 4°C, followed by α-mouse and α-rat secondary abs and counterstaining with DAPI. 3D projections have been created with Imaris-Viewer (v.9.8). The contrast and brightness of representative images (*Figure 1B–C*) remained unaltered, to highlight the detection level of the abs taken up by rLCs. All abs and reagents used in this study are listed in Key resources table.

## Generation of an RTN1A expressing THP-1 cell line

Human RTN1A (NM_0211369) was gene synthesized (Eurofins), cloned into the lentiviral expression vector pHR-SIN-BX-IRES-Emerald (*Paster et al., 2013*) and expressed in the THP-1 cell line (ATCC TIB-202). Following puromycin selection, stable RTN1A protein expression was tested by flow cytometry (α-RTN1A-APC labeled ab, mon161, Novus Biologicals, Biotechne). Furthermore, the authenticity of the THP-1 cell line was confirmed by flow cytometry using a panel of abs to monocytic markers and to HLA-A2. Cell lines were tested for absence of mycoplasma, using a reporter system described recently (*Battin et al., 2017*).

## Differentiation of THP-1 wt and THP-1 RTN1A⁺ cells toward Mφs

THP-1 wt and THP-1 RTN1A⁺ cells were seeded on cover slips in a 24-well plate ($2.5 \times 10^4$ cells/well) and polarized toward Mφs for 72 hr with 50 ng/ml phorbol 12-myristate 13-acetate (Abcam) in RPMI medium and supplements as described previously (*Pinto et al., 2021*).

## Immunofluorescence staining of THP-1 cells and THP-1 Mφs

THP-1 wt and THP-1 RTN1A⁺ cells (both seeded at $2 \times 10^4$) on adhesion slides (Marienfeld) were fixed with acetone for 10 min at RT, stained with α-vimentin and α-RTN1A abs and mounted. For co-localization assays, THP-1 RTN1A⁺ cells were seeded in eight-well chamber slides ($2 \times 10^4$/well, ibidi), coated with 0.1 ng/ml of fibronectin (PromoCell). After 24 hr the cells were washed, fixed with 4% formaldehyde (SAV Liquid Production) for 10 min at RT, and permeabilized with 0.1% Triton X-100 (Sigma-Aldrich) in PBS for 10 min at 4°C. These samples were stained additionally with phalloidin-AF647 probe (F-actin, Invitrogen) for 1 hr at RT. Mφs differentiated on cover slips were fixed and processed as described above for THP-1 RTN1A⁺ cells on coated slides.

## Microscopy and image analysis

A confocal laser scanning microscope (Olympus, FLUOVIEW-FV 3000, equipped with OBIS lasers: 405, 488, 561, 640 nm and ×20, ×40, or ×60 UPlanXApo objectives) and Olympus FV31S-SW software were used in this study. Images were acquired with ×20 objective as Z-stack from four fields of view (FOVs) per epidermal sheet from four different donors and analyzed using ImageJ Fiji software (*Schindelin et al., 2012*). The measurement of the integrated density from the region of interest (ROI) was based on Z-projections with max intensity of manually thresholded images (analogue parameters were used for all analyzed images). Between 100 and 200 ROIs (rLCs) were analyzed per four FOVs.

## Evaluation of the morphology and dendricity of rLCs and THP-1 Mφs

The enumeration of roundish (none or one dendrite) and dendritic (two or more dendrites) rLCs per 0.04 mm² in epidermal sheets was based on vimentin staining and performed using ImageJ Fiji. Sixty to 300 cells were analyzed per FOV. The average length of rLC dendrites and the distance of dendrites from the middle of the cell body was analyzed and quantified using simple neurite tracer (SNT) and Sholl analysis plugin in ImageJ Fiji (*Longair et al., 2011*; *Ferreira et al., 2014*). ×60 objective was used for representative images in *Figure 2A*. The length of cell protrusions in Mp0 was also quantified using SNT (10–40 cells/FOV from four FOVs).

## Analysis of THP-1 and THP-1 Mφ cell areas and co-localization of RTN1A with cytoskeleton structures

To estimate the full size of the cell body of vimentin- and F-actin-stained Mφs on adhesion slides, vimentin and F-actin channels were merged, thresholded, and the ROI area analyzed and quantified using ImageJ Fiji. To assess co-localization of RTN1A with vimentin and F-actin in THP-1 RTN1A+ cells and THP-1 Mφs, we used Manders' coefficient analysis with RTN1A as M1 and vimentin/F-actin as M2. Single Z-stack slices from the bottom, middle, and the top of 10 cells/FOV from four FOVs were analyzed.

## Flow cytometry analysis of eLCs

### Enumeration of eLCs

Cultivation periods and treatments were performed as described above (cultivation of epidermal explants with an α-RTN1A ab) in 96-well round bottom plates. Epidermal explants (6 mm in diameter/1 explant per well in triplicates per condition and time point) were removed and cells in the culture medium were collected from wells (*Figure 3A and B*). Cells were washed with PBS (Gibco, Thermo Fisher Scientific), stained with fixable viability dye and an ab cocktail for LC surface markers (FITC-conjugated CD207 [Beckman Coulter], CD1a [BD Pharmingen]), subsequently fixed, permeabilized, and stained with a secondary α-mouse cross-absorbed F(ab') ab fragment conjugated with AF546 (Thermo Fisher Scientific) for the detection of primary abs. Samples were acquired using FACS Verse (BD Biosciences) and BD Suite software (v1.0.5.3841, BD Biosciences). Viable CD207+CD1a+ eLCs were enumerated (*Figure 3A*) and the signal of the primary ab measured and shown as MFI of AF546 (Thermo Fisher Scientific) (*Figure 3B*).

### Analysis of the eLC phenotype after 24 hr of culture

Cells were processed as described in the previous paragraph and then stained with abs directed against CD1a (BD Pharmingen), CD207 (Beckman Coulter), CCR7 (Miltenyi Biotec), and CD86 (BD Pharmingen). Next, cells were fixed, permeabilized, and stained intracellularly with an APC-conjugated α-RTN1A ab (mon161, Novus Biologicals, Biotechne). The samples were acquired using FACS Verse (BD Biosciences) and BD Suite software (v1.0.5.3841, BD Biosciences). For evaluation, only pre-gated viable CD207+ eLCs were used. Further, doublets and dead cells were excluded. Data was analyzed using the FlowJo software (v10.0.7r2, BD Biosciences). Mean percentages of positive cells and MFI values from triplicates including five donors were analyzed using GraphPad Prism (v8.0.1) (*Figure 3C*).

## Measurement of adhesion molecules, cytokines, and chemokines in epidermal explant and cell culture supernatants

LEGENDplex Human Adhesion Molecule Panel (13-plex) w/VbP LEGENDplex (Biolegend) and human inflammatory panel 1 (13-plex) w/VbP (Biolegend) was used to analyze supernatants from epidermal sheet and cell culture. The assay was carried out according to the manufacturer's instruction. Data analysis was performed using LegendPlex v8.0 software (BioLegend).

## Comparative evaluation of cell aggregate formation by THP-1 wt and THP-1 RTN1A+ cells

Images of THP-1 wt and THP-1 RTN1A+ cells in cultures were taken with the ZOE fluorescence cell imager (Bio-Rad), and the numbers of cellular aggregates per 0.7 mm$^2$ from four FOVs have been quantified using ImageJ Fiji (*Figure 5G*). For enumeration of small and big clusters, an average area of 312.5 and 637.6 µm$^2$ were chosen, respectively (*Figure 4—figure supplement 1*).

## Measurement of cell proliferation and cell number

CellTrace CFSE cell proliferation dye (Invitrogen) was used according to the manufacturer's instructions. Briefly, THP-1 wt and RTN1A+ cells were incubated with CFSE (5 µM) for 20 min at 37°C, washed and seeded in a 24-well plate ($2 \times 10^4$ cells for each condition). Cells were collected at 0, 24, 48 and 72 hr and acquired with FACS Verse. In parallel, the cell number was assessed at the same time points with Trypan blue solution (Sigma) and Neubauer chamber (BRAND).

## Measurement of calcium flux in THP-1 wt and THP-1 RTN1A$^+$ cells

Ratiometric calcium flux experiments with Fura Red (Invitrogen) were performed similar to a previously described method (*Wendt et al., 2015*). Briefly, $1 \times 10^6$ THP-1 wt and RTN1A$^+$ cells were washed, resuspended in 100 µl medium containing 1 µM Fura Red and incubated 30 min at 37°C. Cells were washed once with medium, resuspended in 1 ml medium, and incubated for another 30 min at 37°C. Thereafter, cells were rested on ice until data acquisition at a FACSAria III flow cytometer (BD Bioscience). For the measurement of intracellular calcium flux, 300 µl Fura Red-loaded cells were transferred to FACS tube, pre-warmed for 5 min at 37°C and the baseline response was recorded for 30 s. After adding with 1 µg/ml ionomycin (InvivoGen) or thapsigargin (Invitrogen), cell responses were recorded for 5 min to analyze changes in calcium mobilization. Fura Red was excited using a violet laser (405 nm) and a green laser (561 nm) and changes in emission were detected with a 635LP, 660/20 BP and a 655LP, 795/40 BP filter set, respectively. The ratiometric 'Fura Red Ratio' over time was calculated using the Kinetics tool in FlowJo software version 9.3.3 (Tree Star Inc) as follows:

$$\text{Fura Red Ratio} = \frac{\text{increase of 405 nm induced emission}}{\text{decrease of 561 nm induced emission}}$$

## Statistical analysis

Statistical analysis of the data has been performed using GraphPad Prism (v8.0.1) software. The number of technical and biological replicates have been implicated in respective method sections and figure legends. The statistical tests were adapted to the experimental design: for comparison of two samples (Student's t test), for higher number of samples with replicates (two-way ANOVA with Tukey's or Durrett's multiple-comparison test). In some figures p values were displayed to indicate a tendency, despite lacking significance. Asterisks indicate significant p values; ns – not significant, $p \geq 0.05$, *$p \leq 0.05$, **$p \leq 0.01$, ***$p \leq 0.001$, and ****$p \leq 0.0001$.

## Acknowledgements

We are grateful to all patient-volunteers, who donated material for this study. We thank Christian Schneeberger (Medical University of Vienna, Department of Gynecology) for providing access to the BD FACSVerse instrument. We thank our colleagues from the Department of Dermatology, Medical University of Vienna for their support and scientific feedback. This work was supported by the Austrian Science Fund (P31485-B30) and DK W1248-B30.

## Additional information

### Funding

| Funder | Grant reference number | Author |
| --- | --- | --- |
| Austrian Science Fund | P31485-B30 | Adelheid Elbe-Bürger |
| Austrian Science Fund | DK W1248-B30 | Adelheid Elbe-Bürger |

The funders had no role in study design, data collection and interpretation, or the decision to submit the work for publication.

### Author contributions

Małgorzata Anna Cichoń, Conceptualization, Resources, Formal analysis, Validation, Investigation, Visualization, Methodology, Writing – original draft, Writing – review and editing; Karin Pfisterer, Judith Leitner, Resources, Investigation, Methodology, Writing – review and editing; Lena Wagner, Investigation, Methodology, Writing – review and editing; Clement Staud, Peter Steinberger, Resources, Writing – review and editing; Adelheid Elbe-Bürger, Conceptualization, Resources, Supervision, Funding acquisition, Writing – original draft, Project administration, Writing – review and editing

### Author ORCIDs

Małgorzata Anna Cichoń http://orcid.org/0000-0001-9824-9207
Karin Pfisterer http://orcid.org/0000-0003-1723-6266

Judith Leitner http://orcid.org/0000-0002-7156-1759
Peter Steinberger http://orcid.org/0000-0001-6848-4097
Adelheid Elbe-Bürger http://orcid.org/0000-0003-2461-0367

## Ethics

Human subjects: In this study samples of human skin from healthy volunteers have been used and the processing of the human skin samples is described in the Methods section of the manuscript. In brief, the abdominal skin samples from anonymous healthy donors (female/male, age range: 27-44 years) were obtained during plastic surgery procedures. The study was approved by the local ethics committee of the Medical University of Vienna and conducted in accordance with the principles of the Declaration of Helsinki (ECS 1149/2011 and ECS 1969/2021). Written informed consent was obtained from all participants.

## Decision letter and Author response

Decision letter https://doi.org/10.7554/eLife.80578.sa1

---

# Additional files

## Supplementary files

• Transparent reporting form

## Data availability

All data used, generated, and analyzed during this study are included in the manuscript, Figures 1-7, Methods, and supplementary materials.

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
