## [Editor Report]

This is an important manuscript that establishes a novel link between a novel molecule named Reticulon 1A expressed in Human Langerhans cells and their important immune functions such as migration to lymph nodes and T cell activation. The data presented are novel and convincing. Notably, the authors present intriguing parallels with the mechanisms that control dendrite growth in neurons which may also be relevant for migrating dendritic cells. Altogether, this manuscript will be of wide interest to the scientific community working in the fields of immunology, vaccinology, dermatology, cell biology, and beyond.

---

## [Decision Letter]

**Decision letter after peer review:**

Thank you for submitting your article "Interoperability of RTN1A in dendrite dynamics and immune functions in human Langerhans cells" for consideration by *eLife*. Your article has been reviewed by 2 peer reviewers, and the evaluation has been overseen by a Reviewing Editor and Tadatsugu Taniguchi as the Senior Editor. The following individuals involved in the review of your submission have agreed to reveal their identity: Clare Bennett (Reviewer #2).

Essential revisions:

Essential revisions are textual and for clarity as suggested by the comments of the reviewers below. There is no need for additional experiments suggested in points number 4d,e,g from Reviewer #1, except if the authors can add them.

*Reviewer #1 (Recommendations for the authors):*

Some comments and suggestions are listed here. The majority of comments and suggestions relate to helpful clarifications of experimental issues and better readability and comprehensibility. A few simple experiments, that would not need any new method or resources (other than human skin) may enhance the authors' conclusions. Such experiments are suggested in points number 4d,e,g.

1. Materials & Methods. The authors write "eLCs after ab uptake were collected and analysed for the presence of primary abs by staining of viable eLCs with secondary α-mouse cross-absorbed F(ab') antibody fragment conjugated with AF546 (Thermofisher Scientific), followed by CD1a…" For their regular RTN1A staining the authors permeabilised the cells, as mentioned above in the cited sentence. Wouldn't the detection of antibody uptake over the different time periods in culture also need permeabilisiation? Please clarify.

2. Figure 1 and Supplemental Figure 1. The authors write: "Notably, only RTN1A+ rLCs captured the RTN1A ab (Figure 1C, yellow arrows). Indeed, not all rLCs and eLCs express RTN1A. We analysed the frequency of RTN1A expression in human primary LCs and found that around 80% of rLCs and eLCs express this protein (S. Figure 1A), from which ~70% of all rLCs and eLCs co-express CD207 and CD1a, respectively (S. Figure 1B,C)." It is not clear to me how the authors draw the conclusion for the first sentence above. I panel 1C they stain with the secondary anti-mouse Ig for the captured anti-RTN1A, and with anti-Langerin for the LCs. How do we know in this setting that the RTN1A-uptake+ LCs express RTN1A? Theoretically, this must be so. Otherwise, where would the anti-RTN1A in the medium have bound to? Nevertheless, this conclusion seems formally not correct to me. Would such a modified wording of the paragraph be more understandable and – above all – correct?: "Resident LCs readily captured the RTN1A ab (Figure 1C, yellow arrows). Notably, not all rLCs did so (Figure 1C, white arrows). Indeed, not all rLCs and eLCs express RTN1A to which the antibody could have bound during the culture period, as shown by the analysis of the frequency of RTN1A expression (not uptake!) in suspensions of human primary LCs: We found that around 80% of rLCs and eLCs express RTN1A (S. Figure 1A). ???from which ~70% of all rLCs and eLCs co-express CD207 and CD1a, respectively (S. Figure 1B, C)." Please make this issue clear. The very last sentence is difficult to understand. Aren't LCs in panel S1A already defined on the basis of CD207 (or CD1a)? Is the "co-expression" of CD207 and CD1a based on a calculation? I do not see a double-staining. Obviously, this will be the case – no doubt. Co-expression of these markers has been often shown. Nevertheless, the description of these – convincing – data should be made clearer.

3. Figure 2C. "Sholl analyses". The authors write: "Sholl analysis addressing frequency and complexity of dendrite brunches (TYPO!!) showed significantly decreased rLC dendrite distribution in α-RTN1A ab-treated epidermal sheets in comparison to controls and most pronounced to UT epidermal sheets (Figure 2C)." However, the authors do not indicate a significance between controls (i.e., isotype) and RTN1A. At first glance, it, therefore, looks as if there were no difference between isotype and anti-RTN1A. Please clarify.

4. Data from Figure 3 – some points that need clarification in the text or in M&M or in the legend.

4a. Figure 3A. Are the numbers of emigrated LC per 6mm explant or per triplicate 96-wells, i.e., 3 explants? Please state explicitly.

4b. Figure 3A. Are these cell numbers based on direct cell counts in the hemocytometer correlated with the percentage of CD207? or MHC-II? positive cells as determined by FACS? At 24h it would also be possible to directly count "hairy" cells, i.e., LC. Please specify.

4c. Figure 3B. Explants had been cultured for the indicated times in the presence of anti-RTN1A or an isotype control – or left untreated. Cells retrieved from the wells were then stained with a fluorochrome-conjugated secondary anti-mouse-Ig. Signals can be detected for the isotype as well. Does this mean that after 3h of incubation/culture, the signal is not RTN1A-specific? Only after 6 and, even more so, after 24h the increase in RTN1A signal reflects true antigen-specific uptake? Figures1B,C would suggest so. Please make this clear.

4d. Another means that could confirm and buttress the observations from 3A would be a correlation with LCs that are left behind in the sheets – obviously, not at the early time points but rather at 48 and/or 72h of culture in the presence of anti-RTN1A / isotype. In fact, the evaluation of cell numbers in the experiments depicted in Figure 2 should answer this question for the 24h time point.

4e. Figure 3A and B. Most human (and mouse as well) skin explant cultures run over 2-3 days. It may be informative and strengthen the thrust of the manuscript, if these two time points were also investigated, especially with regard to the simple cell numbers (panel A).

4f. Figure 3C. FACS analyses. Does "%of" mean percent of all emigrés or was any pre-gating (MHC-II? Langerin?) applied. Please specify.

4g. Figure 3C. FACS analyses. When sheets are cultured in the presence of anti-RTN1A, CCR7 on LCs goes up, CD86 goes down. Both are maturation markers, therefore, this seems a bit strange at first glance. But it is what it is… Nevertheless, the authors are encouraged to add to this panel also CD83, another classical DC/LC maturation marker, actually used in other experiments of this manuscript anyway.

5. Could there be any link to the old observations of Ralf Ross from Mainz with "fascin", the actin-bundling protein, expressed in mature Langerhans cells / dendritic cells? Maybe worth a line or two of Discussion?

*Reviewer #2 (Recommendations for the authors):*

I have the following suggestions/questions:

1. "Sholl analysis" is used for Figure 2B. It would help if this is explained in the text.

2. For figure 3D the authors use the Legendplex system with supernatants from total epidermal cell cultures to draw conclusions about the expression of cell adhesion molecules. From this approach, it is not clear which cells the proteins are expressed on and therefore whether the effect on LC residency is direct or indirect. This is important because the downregulation of CD44, PCAM-1, and NCAM are highlighted in the graphical abstract. Do the authors have any flow data for e.g. EpCAM expression on the LC that could add to this information? And/or this caveat should be mentioned in the discussion. It would also be good to see some of the flow data for the phenotypic changes shown in 3A.

3. Did the authors investigate the role of RTN1A in Thp1 cells that had been skewed to an LC-like phenotype by the addition of TGFb and other factors? It would be interesting to understand where RTN1A stands in the hierarchy in signals between TGFb-controlled residency and PAMP-induced activation and migration.

---

## [Author Response]

Reviewer #1 (Recommendations for the authors):

Some comments and suggestions are listed here. The majority of comments and suggestions relate to helpful clarifications of experimental issues and better readability and comprehensibility. A few simple experiments, that would not need any new method or resources (other than human skin) may enhance the authors' conclusions. Such experiments are suggested in points number 4d,e,g.

1. Materials & Methods. The authors write "eLCs after ab uptake were collected and analysed for the presence of primary abs by staining of viable eLCs with secondary α-mouse cross-absorbed F(ab') antibody fragment conjugated with AF546 (Thermofisher Scientific), followed by CD1a…" For their regular RTN1A staining the authors permeabilised the cells, as mentioned above in the cited sentence. Wouldn't the detection of antibody uptake over the different time periods in culture also need permeabilisiation? Please clarify.

Indeed, we described this only once because we thought that it is obvious that detection of intracellular molecules such as RTN1A or captured abs against RTN1A in LCs always require permeabilization. To avoid confusion we have now implemented this information wherever relevant in Materials and methods (page 23).

2. Figure 1 and Supplemental Figure 1. The authors write: "Notably, only RTN1A+ rLCs captured the RTN1A ab (Figure 1C, yellow arrows). Indeed, not all rLCs and eLCs express RTN1A. We analysed the frequency of RTN1A expression in human primary LCs and found that around 80% of rLCs and eLCs express this protein (S. Figure 1A), from which ~70% of all rLCs and eLCs co-express CD207 and CD1a, respectively (S. Figure 1B,C)." It is not clear to me how the authors draw the conclusion for the first sentence above. I panel 1C they stain with the secondary anti-mouse Ig for the captured anti-RTN1A, and with anti-Langerin for the LCs. How do we know in this setting that the RTN1A-uptake+ LCs express RTN1A? Theoretically, this must be so. Otherwise, where would the anti-RTN1A in the medium have bound to? Nevertheless, this conclusion seems formally not correct to me. Would such a modified wording of the paragraph be more understandable and – above all – correct?: "Resident LCs readily captured the RTN1A ab (Figure 1C, yellow arrows). Notably, not all rLCs did so (Figure 1C, white arrows). Indeed, not all rLCs and eLCs express RTN1A to which the antibody could have bound during the culture period, as shown by the analysis of the frequency of RTN1A expression (not uptake!) in suspensions of human primary LCs: We found that around 80% of rLCs and eLCs express RTN1A (S. Figure 1A). ???from which ~70% of all rLCs and eLCs co-express CD207 and CD1a, respectively (S. Figure 1B, C)." Please make this issue clear. The very last sentence is difficult to understand. Aren't LCs in panel S1A already defined on the basis of CD207 (or CD1a)? Is the "co-expression" of CD207 and CD1a based on a calculation? I do not see a double-staining. Obviously, this will be the case – no doubt. Co-expression of these markers has been often shown. Nevertheless, the description of these – convincing – data should be made clearer.

We thank the Reviewer to have unraveled this inaccuracy of our description and appreciate his excellent suggestion for improvement. We have implemented his recommendation with minor modifications necessary for further clarification (page 6). The information how and according to which markers LCs were gated was missing and was now implemented in the Figure Legend (Figure 1—figure supplement).

3. Figure 2C. "Sholl analyses". The authors write: "Sholl analysis addressing frequency and complexity of dendrite brunches (TYPO!!) showed significantly decreased rLC dendrite distribution in α-RTN1A ab-treated epidermal sheets in comparison to controls and most pronounced to UT epidermal sheets (Figure 2C)." However, the authors do not indicate a significance between controls (i.e., isotype) and RTN1A. At first glance, it, therefore, looks as if there were no difference between isotype and anti-RTN1A. Please clarify.

The remark that the difference in the frequency and complexity of LC dendrite distribution between the isotype and α-RTN1A ab-treated epidermal sheets is not significant is correct. Not significant (ns) data were not labeled throughout the Figures due to limited space above the bar graphs but have now included it in the revised Figure 2C for clarity. As a significant difference was observed between LC dendrite frequency and complexity in untreated (UT) and isotype-treated epidermal sheets and LCs in UT and α-RTN1A ab-treated epidermal sheets only, we concluded that the inhibition of RTN1A rather affected the length of the dendrites but not their distribution. This result has now been better explained in the revised manuscript (page 7).

4. Data from Figure 3 – some points that need clarification in the text or in M&M or in the legend.

4a. Figure 3A. Are the numbers of emigrated LC per 6mm explant or per triplicate 96-wells, i.e., 3 explants? Please state explicitly.

We are grateful to the reviewer for bringing this point up and have added a more comprehensive information in the revised Materials and methods section (page 23).

4b. Figure 3A. Are these cell numbers based on direct cell counts in the hemocytometer correlated with the percentage of CD207? or MHC-II? positive cells as determined by FACS? At 24h it would also be possible to directly count "hairy" cells, i.e., LC. Please specify.

We have now described in Materials and methods how cell numbers were exactly evaluated (page 23).

4c. Figure 3B. Explants had been cultured for the indicated times in the presence of anti-RTN1A or an isotype control – or left untreated. Cells retrieved from the wells were then stained with a fluorochrome-conjugated secondary anti-mouse-Ig. Signals can be detected for the isotype as well. Does this mean that after 3h of incubation/culture, the signal is not RTN1A-specific? Only after 6 and, even more so, after 24h the increase in RTN1A signal reflects true antigen-specific uptake? Figures1B,C would suggest so. Please make this clear.

The point of the reviewer is well taken. Our data show undoubtedly that both abs have been captured by LCs after three hours of incubation. However, it is currently unclear to which extent data at this time point of evaluation mirror ab specificity which is unambiguously the case at later time points as accurately mentioned by the reviewer.

4e. Figure 3A and B. Most human (and mouse as well) skin explant cultures run over 2-3 days. It may be informative and strengthen the thrust of the manuscript, if these two time points were also investigated, especially with regard to the simple cell numbers (panel A).

4d. Another means that could confirm and buttress the observations from 3A would be a correlation with LCs that are left behind in the sheets – obviously, not at the early time points but rather at 48 and/or 72h of culture in the presence of anti-RTN1A / isotype. In fact, the evaluation of cell numbers in the experiments depicted in Figure 2 should answer this question for the 24h time point.

These are great ideas and we have indeed planned to address them in adequate experiments. In fact, it was the reason why we have extended our manuscript resubmission to the end of the months hoping to receive skin. However, surgeries are generally extremely rare during the summer season in our hospital due to bad wound healing. We were not fortunate enough to receive skin this month.

4f. Figure 3C. FACS analyses. Does "%of" mean percent of all emigrés or was any pre-gating (MHC-II? Langerin?) applied. Please specify.

Yes, FACS data about the expression of eLC markers in Figure 3C derive from pre-gated CD207^+^ eLCs. This information has been implemented in the revised Materials and methods section (page 23) as well as in the respective Figure legend.

4g. Figure 3C. FACS analyses. When sheets are cultured in the presence of anti-RTN1A, CCR7 on LCs goes up, CD86 goes down. Both are maturation markers, therefore, this seems a bit strange at first glance. But it is what it is… Nevertheless, the authors are encouraged to add to this panel also CD83, another classical DC/LC maturation marker, actually used in other experiments of this manuscript anyway.

As we have not received skin for experiments we were not able to include this marker.

5. Could there be any link to the old observations of Ralf Ross from Mainz with "fascin", the actin-bundling protein, expressed in mature Langerhans cells / dendritic cells? Maybe worth a line or two of Discussion?

We are grateful for bringing this important literature to our attention. In fact, we could identify some similarities between our studies such as cell type-specific expression of the molecule of interest [fascin (F-actin) and RTN1A (endoplasmic reticulum)] and their crosstalk with the cytoskeleton. Parts of our study are in line with their attempt to investigate the dynamics of cytoskeletal remodeling in professional antigen-presenting cells via inhibition experiments. Although RTN1A is not a component of the cytoskeleton like fascin, we similarly demonstrated its involvement regulating cellular protrusion. In contrast to the well described genetic regulation of fascin, this is unknown for RTN1A in professional antigen-presenting cells and needs to be investigated in the future. As recommended, we have revised the discussion accordingly (page 13).

Reviewer #2 (Recommendations for the authors):

I have the following suggestions/questions:

1. "Sholl analysis" is used for Figure 2B. It would help if this is explained in the text.

The requested explanation has now been inserted in the revised manuscript (page 7). Notably, we have used an automated Sholl analysis (Ferreira et al. 2014, DOI: 10.1038/nmeth.3125) and is now supplemented in the revised manuscript in Materials and methods (page 22).

2. For figure 3D the authors use the Legendplex system with supernatants from total epidermal cell cultures to draw conclusions about the expression of cell adhesion molecules. From this approach, it is not clear which cells the proteins are expressed on and therefore whether the effect on LC residency is direct or indirect. This is important because the downregulation of CD44, PCAM-1, and NCAM are highlighted in the graphical abstract. Do the authors have any flow data for e.g. EpCAM expression on the LC that could add to this information? And/or this caveat should be mentioned in the discussion.

Legendplex is a bead-based immunoassay that enables detection of free or cell-bound molecules in cell culture supernatants via their binding to specific detection beads and their subsequent analysis using pre-defined mixture of antibodies thus using the principles of sandwich ELISAs. Because of the broad molecule panel and assembly of this assay it is not possible to assign molecules of interest such as EpCAM or CD44 to a particular cell type. This information was added in the result section of the revised manuscript (page 8-9). While we cannot specify which cells expressed the measured adhesion molecules, it is published that some adhesion molecules can be enzymatically cleaved during the process of tissue remodeling and their extracellular components can be detected (DOI: 10.1371/journal.pone.0029305). Even though we do not know the exact mechanism by which RTN1A controls the dendrites of LCs, our Legendplex data show that the inhibition of RTN1A in rLCs reduces adhesion molecule levels in culture medium, implying a direct effect on tissue remodeling.

When analyzing LC markers such as CCR7 for cell migration and CD86 for cell activation (Figure 3C), we did not investigate expression of selected adhesion molecules, because at that stage we were not aware that the consequences of RTN1A inhibition will be associated with changes in cell adhesion.

It would also be good to see some of the flow data for the phenotypic changes shown in 3A.

The analysis shown in Figure 3A was based on staining of viable eLCs stained for CD1a and CD207 and subsequent FACS analysis and presentation of eLCs numbers in bar graphs. As requested, we provide additional representative dot-blots for the LC phenotype at 24 h for your information only. As no remarkable differences were observed we have not included these data in the manuscript (see Author response image 1).

**Author response image 1. sa2fig1:** 

3. Did the authors investigate the role of RTN1A in Thp1 cells that had been skewed to an LC-like phenotype by the addition of TGFb and other factors? It would be interesting to understand where RTN1A stands in the hierarchy in signals between TGFb-controlled residency and PAMP-induced activation and migration.

It was not the scope of our study to investigate a potential crosstalk between the TGF-b signaling pathway and the effect of RTN1A on LC morphology and have therefore not established the differentiation procedure and subsequent analysis. As this could be perhaps translated to the behavior of primary LCs we would be interested in addressing this question in future studies. We appreciate this thought and suggestion.